# Neural Deconstruction Search for Vehicle Routing Problems

**André Hottung** *andre.hottung@uni-bielefeld.de*
*Bielefeld University, Germany*

**Paula Wong-Chung** *paula.wong-chung@alumni.ubc.ca*
*University of British Columbia, Canada*

**Kevin Tierney** *kevin.tierney@uni-bielefeld.de*
*Bielefeld University, Germany*

**Reviewed on OpenReview:** *https://openreview.net/forum?id=bCmEP1Ltwq*

## Abstract

Autoregressive construction approaches generate solutions to vehicle routing problems in a step-by-step fashion, leading to high-quality solutions that are nearing the performance achieved by handcrafted operations research techniques. In this work, we challenge the conventional paradigm of sequential solution construction and introduce an iterative search framework where solutions are instead *deconstructed* by a neural policy. Throughout the search, the neural policy collaborates with a simple greedy insertion algorithm to rebuild the deconstructed solutions. Our approach matches or surpasses the performance of state-of-the-art operations research methods across three challenging vehicle routing problems of various problem sizes.

## 1 Introduction

Methods that can learn to solve complex optimization problems have the potential to transform decision-making processes across virtually all domains. It is therefore unsurprising that learning-based optimization approaches have garnered significant attention and yielded substantial advancements (Bello et al., 2016; Kool et al., 2019; Kwon et al., 2020). Notably, reinforcement learning (RL) approaches are particularly promising because they do not rely on a pre-defined training set of representative solutions and can develop new strategies from scratch for novel optimization problems. These methods generally construct solutions *incrementally* through a sequential decision-making process and have been successfully applied to various vehicle routing problems.

Despite recent progress, learning-based methods for combinatorial optimization (CO) problems usually fall short of outperforming the state-of-the-art techniques from the operations research (OR) community. For instance, while some new construction approaches for the capacitated vehicle routing problem (CVRP) have surpassed the LKH3 solver (Helsgaun, 2000), they still struggle to match the performance of the state-of-the-art HGS solver (Vidal et al., 2012), particularly for larger instances with over 100 nodes. One reason for this is their inability to explore as many solutions as traditional approaches within the same amount of time. Given the limitations of current construction approaches, we propose challenging the traditional paradigm of sequential solution construction by introducing a novel iterative search framework, *neural deconstruction search (NDS)*, which instead deconstructs solutions using a neural policy.

NDS is an iterative search method designed to enhance a given solution through a two-phase process involving deconstruction and reconstruction along the lines of large neighborhood search (LNS) (Shaw, 1998) and ruin-and-recreate (Schrimpf et al., 2000) paradigms. The deconstruction phase employs a deep neural network (DNN) to determine the customers to be removed from the tours of the current solution. This is achieved through a sequential decision-making process, in which nodes are removed one at a time based on the network's guidance. The reconstruction phase utilizes a straightforward greedy insertion algorithm, which

inserts customers in the order given by the neural network at the locally optimal positions. Note that NDS is trained using reinforcement learning, which makes it adaptable to problems for which no reference solutions are available for training.

The overall concept of modifying a solution by first removing some solution components and then conducting a rebuilding step has been successfully used in various vehicle routing problem methods. Non-learning-based methods that use this concept include the rip-up and reroute method from Dees & Smith (1981), LNS from Shaw (1998), and the ruin and recreate method from Schrimpf et al. (2000). Learning-based methods have also harnessed this paradigm. The local rewriting method from Chen & Tian (2019), neural large neighborhood search from Hottung & Tierney (2020), and the random reconstruction technique introduced in Luo et al. (2023) employ a DNN during the reconstruction phase. The approaches from Li et al. (2021) and Falkner & Schmidt-Thieme (2023) both generate different subproblems for a given solution and then use a DNN to choose which subproblem should be considered in the reconstruction phase.

NDS has been designed with the goal of achieving a fast search procedure without sacrificing the high-quality search guidance of a DNN. For medium-sized CVRP instances with 500 customers, state-of-the-art OR approaches such as SISRs (Christiaens & Vanden Berghe, 2020) can examine upwards of 270k solutions per second, however neural combinatorial optimization approaches, like POMO (Kwon et al., 2020), can only create around 10k per second. In contrast, NDS can process 120k solutions per second, significantly more than existing neural construction techniques. When combined with a powerful deconstruction DNN, NDS is able to outperform state-of-the-art OR approaches like SISRs and HGS in similar wall-clock time.

We evaluate NDS on several challenging problems, including the CVRP, the vehicle routing problem with time windows (VRPTW), and the prize-collecting vehicle routing problem (PCVRP). NDS demonstrates substantial performance gains compared to existing learned construction methods and matches or surpasses state-of-the-art OR methods across various routing problems of different sizes. To the best of our knowledge, NDS is the first learning-based approach that achieves this milestone.

In summary, we provide the following contributions:

- We propose to use a learned deconstruction policy in combination with a simple greedy insertion algorithm.

- We introduce a novel training procedure designed to learn effective deconstruction policies.

- We present a new network architecture optimized for encoding the current solution.

- We develop a high-performance search algorithm specifically designed to leverage the parallel computing capabilities of GPUs.

## 2 Literature Review

**Construction Methods** The introduction of the pointer network architecture by Vinyals et al. (2015) marked the first autoregressive, deep learning-based approach for solving routing problems. In their initial work, the authors employ supervised learning to train the models, demonstrating its application to the traveling salesperson problem (TSP) with 50 nodes. Building on this, Bello et al. (2016) propose using reinforcement learning to train pointer networks, showcasing its effectiveness in addressing larger TSP instances.

For the more complex CVRP, the first learning-based construction methods were introduced by Nazari et al. (2018) and Kool et al. (2019). Recognizing the symmetries inherent in many combinatorial optimization problems, Kwon et al. (2020) develop POMO, a method that leverages these symmetries to improve exploration of the solution space during both training and testing. Extending this concept, Kim et al. (2022) propose a general-purpose symmetric learning framework.

Various techniques have been proposed to enhance performance in neural combinatorial optimization. For instance, Hottung et al. (2022) introduce efficient active search, which updates a subset of model parameters during inference. Choo et al. (2022) propose SGBS, combining Monte Carlo tree search with beam search

to guide the search process more effectively. Additionally, Drakulic et al. (2023) and Luo et al. (2023) focus on improving out-of-distribution generalization by re-encoding the remaining subproblem after each construction step. To enhance solution diversity during sampling, Grinsztajn et al. (2023) and Hottung et al. (2025) explore approaches that learn a set of policies, rather than a single policy.

Instead of constructing solutions autoregressively, some approaches predict heat maps that highlight promising solution components (e.g., arcs in a graph), which are then used in post-hoc searches to construct solutions (Joshi et al., 2019; Fu et al., 2021; Kool et al., 2022b; Min et al., 2023). Other approaches focus on more complex variants of routing problems, such as the VRPTW (Falkner & Schmidt-Thieme, 2020; Kool et al., 2022a; Berto et al., 2024b;c), or the min-max heterogeneous CVRP (Berto et al., 2024a).

**Improvement Methods**  Improvement methods focus on iteratively refining a given starting solution. In addition to the ruin-and-recreate approaches discussed in the introduction, several other methods aim to enhance solution quality through iterative adjustments. For instance, Ma et al. (2021) propose learning to iteratively improve solutions by performing local modifications. Similarly, several works have guided the $k$-opt heuristic for vehicle routing problems (Wu et al., 2019; da Costa et al., 2020), although they are constrained by a fixed, small $k$. More recently, Ma et al. (2023) introduced a method capable of handling any $k$. Furthermore, Ye et al. (2024a) and Kim et al. (2024) integrate learning-based approaches with ant colony optimization to allow for a more extensive search phase. Additionally, several divide-and-conquer methods have been developed to address large-scale routing problems (Kim et al., 2021; Li et al., 2021; Ye et al., 2024b; Zheng et al., 2024).

## 3   Vehicle Routing Problems

Vehicle routing problems (VRPs) represent a broad class of combinatorial optimization problems that are fundamental in logistics and transportation. These problems involve determining optimal routes for a fleet of vehicles to serve a set of customers while satisfying specific constraints. The standard VRP generalizes the well-known TSP and is also NP-hard. In this paper, we consider three key VRP variants: the CVRP, the VRPTW, and the PCVRP. Each variant introduces additional constraints and objectives, making them suitable for different real-world applications. We briefly introduce all three variants below, while a more detailed discussion can be found in Toth & Vigo (2014).

**Capacitated Vehicle Routing Problem**  The CVRP is one of the most studied VRP variants, where each customer has a specific demand that must be met by a fleet of vehicles with limited capacity. Formally, the problem is defined on a complete graph where a depot and a set of $N$ customers, denoted as $c_1, \ldots, c_N$, are represented as nodes with associated coordinates in a two-dimensional Euclidean space. Each customer $c_i$ has a demand $q_i$, and each vehicle has a maximum capacity $Q$. The objective is to determine a set of routes that collectively serve all customers while minimizing the total travel cost. We calculate the travel costs as the sum of Euclidean distances between visited nodes. Each route must start and end at the depot, and the sum of customer demands on any route must not exceed $Q$. An instance of the problem is denoted by $l$, which encapsulates the locations of the depot and customers, their demands, and the vehicle capacity. A solution $s$ for an instance $l$ consists of a set of routes, where each route defines the sequence of customer visits assigned to one vehicle.

**Vehicle Routing Problem with Time Windows**  The VRPTW extends the CVRP by incorporating time constraints on customer deliveries. Each customer $c_i$ is assigned a time window $[a_i, b_i]$, specifying the earliest and latest allowable delivery times. A vehicle may arrive early but must wait until the time window opens. Additionally, each customer has a service time, representing the duration required to complete the delivery before the vehicle can proceed. The objective is to minimize total travel time while ensuring that all deliveries occur within the specified time windows. In this paper, the travel time between two nodes is identical to the Euclidean distance between those nodes. Like the CVRP, all routes start and end at the central depot, and vehicle capacity constraints must be respected.

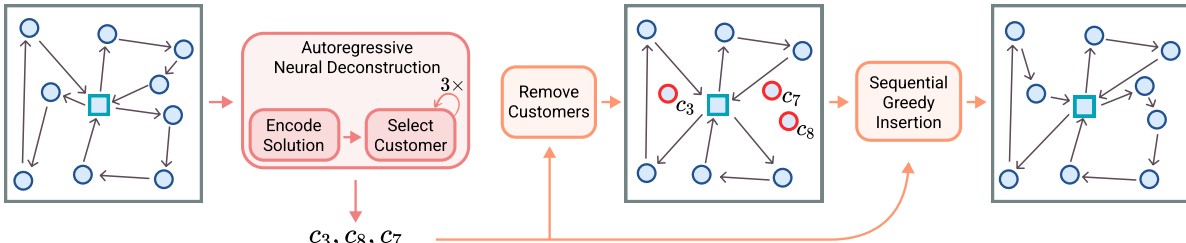

Figure 1: Improving a solution via neural deconstruction.

**Prize-Collecting Vehicle Routing Problem**  The PCVRP is a variant of the VRP in which not all customers need to be visited. Unlike the CVRP and VRPTW, where all customers must be served, the PCVRP assigns a prize to each customer, and the objective is to maximize the difference between the total prize collected and the travel costs. Vehicles still start and end at a central depot and must respect capacity constraints. This variant is particularly useful in scenarios where serving every customer is not mandatory but must be balanced against operational costs.

## 4  Neural Deconstruction Search

NDS is an improvement method for vehicle routing problems. During the search, new candidate solution are generated by applying a neural deconstruction policy followed by a greedy reconstruction algorithm to the current solution. An example for generating a new candidate solution for the CVRP is shown in Figure 1. First, the static instance information $l$ and the current solution $s$ are given to a neural policy which then selects customers for removal from the tours of $s$ in a sequential decision making process. The selected customers are then removed from the tours of $s$, resulting in a (partial) solution in which the selected customers are not part of any tour and are thus unvisited. Finally, the selected customers are reinserted into tours using a simple sequential greedy insertion algorithm that inserts customers in the order defined by the policy.

Note that the number of decision steps (and hence the number of model calls) depends solely on the number of customers that are removed. Since we keep that number fixed across different instances sizes, NDS is able to generate candidate solutions much faster than construction approaches that build solutions from scratch. Furthermore, NDS is built for efficient GPU and CPU interaction by using batched rollouts.

One key requirement for the learned policies is that they contain a sufficient degree of randomness during the deconstruction process. During the search, a good solution is often deconstructed hundreds of times before an improving solution is found. It is hence vital that the learned policy is able to produce multiple diverse deconstruction instructions for a single solution. We encourage diverse output generation by providing the policy with an additional seed input $v$ as proposed in Hottung et al. (2025). During training, the model learns to condition its output on $v$, resulting in the selection of different customer sets for different seed values at test time.

### 4.1  Deconstruction Policy

For solution deconstruction, a neural policy is employed to sequentially select customers for removal from a given solution. We model this selection process as a Markov decision process. Given a solution $s$ for a VRP instance $l$, a policy network $\pi_\theta$, parameterized by $\theta$, is used to select $M$ customers for removal. At each step $m \in \{1, \ldots, M\}$, an action $a_m \in \{1, \ldots, N\}$ is chosen according to the probability distribution $\pi_\theta(a_m \mid l, s, v, a_{1:m-1})$, where $a_m$ corresponds to selecting customer $c_{a_m}$, $v$ is a random seed, and $a_{1:m-1}$ are the previous actions. We condition the policy on a random seed $v$ to encourage more diverse rollouts as explained in Hottung et al. (2025). Each seed is a randomly generated binary vectors of dimension $d_v$ (we set $d_v = 10$ in all experiments). Finally, after all $M$ customers are selected the reward can be computed as discussed in the following sections.

## 4.2 Training

The deconstruction policy in NDS is trained using reinforcement learning. During the training process, solutions are repeatedly deconstructed and reconstructed, aiming to discover a deconstruction policy that facilitates the reconstruction of high-quality solutions. Algorithm 1 outlines our training procedure. It is important to implement the algorithm in a way that allows processing batches of instances in parallel to ensure efficient training. However, for clarity, the pseudocode presented describes the training process for a single instance at a time.

---

**Algorithm 1** NDS Training

1: **procedure** TRAIN(Iterations per instance $I$, rollouts per solution $K$, improvement steps $J$)
2:     Initialize policy network $\pi_\theta$
3:     **while** Termination criteria not reached **do**
4:         $l \leftarrow$ GENERATEINSTANCE( )
5:         $s \leftarrow$ GENERATESTARTSOLUTION($l$)
6:         **for** $j = 1, \ldots, J$ **do**
7:             $s \leftarrow$ IMPROVEMENTSTEP($s, \pi_\theta$)                                          ▷ Improve solution using procedure shown in Figure 2
8:         **end for**
9:         **for** $i = 1, \ldots, I$ **do**
10:            $\{\tau_1, \tau_2, \ldots, \tau_K\} \leftarrow$ ROLLOUTPOLICY($\pi_\theta, l, s, K$)                          ▷ Sample $K$ rollouts
11:            $\bar{s}_k \leftarrow$ REMOVECUSTOMERS($s, \tau_k$)        $\forall k \in \{1, \ldots, K\}$
12:            $s'_k \leftarrow$ GREEDYINSERTION($\bar{s}_k, \tau_k$)        $\forall k \in \{1, \ldots, K\}$
13:            $r_k \leftarrow \max(\text{OBJ}(s) - \text{OBJ}(s'_k), 0)$        $\forall k \in \{1, \ldots, K\}$                          ▷ Calculate reward
14:            $b \leftarrow \frac{1}{K} \sum_{k=1}^{K} r_k$                          ▷ Calculate baseline
15:            $k^* = \arg\max_{k \in \{1, \ldots, K\}} r_k$
16:            $g_i \leftarrow (r_{k^*} - b) \nabla_\theta \log \pi_\theta(\tau_{k^*} | l, s, v_{k^*})$                          ▷ Calculate gradients
17:            $s \leftarrow s'_{k^*}$                          ▷ Update $s$ with best found solution
18:         **end for**
19:         $\theta \leftarrow \theta + \alpha \sum_{i=1}^{I} g_i$                          ▷ Optimizer step with accumulated gradients
20:     **end while**
21: **end procedure**

---

The main training loop runs until a termination criterion (such as the number of processed instances) is met. In each iteration of the loop, a new instance and its corresponding solution are generated in lines 4-8. The solution is then repeatedly deconstructed and reconstructed for $I$ iterations (lines 9-18), during which gradients are computed based on the rewards obtained. After completing $I$ iterations, the gradients are accumulated, and the network parameters are updated using the learning rate $\alpha$. The following section provides a more detailed explanation of this process.

At the start of each iteration of the training loop, a new instance $l$ and its corresponding solution $s$ are generated. The instance is sampled from the same distribution as the test instances. In line 5, an initial solution is constructed using a simple procedure: for an instance with $N$ customers, we generate $N$ tours, each containing one customer. In lines 6-8, this initial solution is iteratively improved through $J$ improvement steps of the NDS search procedure. This search procedure is detailed in Section 4.4. By improving $s$ before the training rollouts, we ensure that the training focuses on non-trivial solutions.

In lines 9 to 18, the solution $s$ is improved over $I$ iterations. At the start of each iteration, the policy $\pi_\theta$ is used to sample $K$ rollouts $\tau_1, \tau_2, \ldots, \tau_K$, using $K$ different, random seed vectors $v_0, \ldots, v_k$. Each rollout is a sequence of $M$ actions that specifies the indices of customers to be removed from the tours in solution $s$. Each rollout $\tau_k$ is individually applied to deconstruct solution $s$ by removing the specified customers, yielding $K$ deconstructed solutions $\bar{s}_1, \ldots, \bar{s}_K$. These deconstructed solutions are then repaired using the greedy insertion algorithm, which is described in more detail below. Next, the reward $r_k$ is calculated for each rollout $\tau_k$, based on the difference in cost between the original solution $s$ and the reconstructed solution $s'_k$. Importantly, the reward is constrained to be non-negative, encouraging the learning of risk-taking policies. In lines 14 to 16, the gradients are computed using the REINFORCE method. The overall probability of generating a rollout $\tau_k$ is given by $\pi_\theta(\tau_k | l, s, v_k) = \prod_{m=1}^{M} \pi_\theta(a_m | l, s, v_k, a_{1:m-1})$. The baseline $b$ is set as the average cost of all rollouts. Gradients are only calculated with respect to the best-performing rollout, denoted $k^*$, to encourage diversity in the solutions as proposed by Grinsztajn et al. (2023). Finally, at the end of each iteration, the solution $s$ is replaced by the reconstructed solution with the highest reward.

**Greedy Insertion** The greedy insertion procedure reintegrates the customers removed by the policy, inserting them one by one into either existing or new tours. Specifically, if $M$ customers have been removed, the procedure performs $M$ iterations, where in each iteration, a single customer $c_{a_m}$ is inserted. At each iteration $m$, the cost of inserting customer $c_{a_m}$ at every feasible position in all tours is evaluated. Throughout this process, various constraints, such as vehicle capacity limits, are taken into account. If at least one feasible insertion point is found within any of the existing tours, the customer $c_{a_m}$ is placed at the position that incurs the least additional cost. If no feasible insertion is available, a new tour is created for customer $c_{a_m}$.

The order in which removed customers are reinserted significantly impacts the overall performance. We reinsert customers either in the order determined by the neural network or at random. Allowing the network to control the reinsertion order gives it control over the reconstruction process, enabling it to find ordering strategies that lead to better reconstructed solutions. If customers are ordered at random, a deconstructed solution should be reconstructed multiple times using different insertion orders. This can provide a more stable learning signal during training.

### 4.3 Model Architecture

We design a transformer-based architecture that consists of an encoder and a decoder. The encoder is used to generate embeddings for all nodes based on the instance $l$ and the current solution $s$. The decoder is used to decode a sequence of actions based on these embeddings in an iterative fashion.

#### 4.3.1 Encoder

The encoder processes the static node features $x_i$ for each of the $N + 1$ nodes in an instance $l$, where $x_0$ corresponds to the depot and $x_1, \ldots, x_N$ correspond to customer nodes. Additionally, it incorporates the current solution $s$. For the CVRP, the depot feature vector $x_0$ includes the depot coordinates, while the customer feature vectors $x_i$ $(i \geq 1)$ include the customer coordinates and demands $q_i$. For the VRPTW, the customer feature vectors are further extended to include the earliest $a_i$ and latest arrival times $b_i$, as well as the service duration. For the PCVRP, the CVRP customer feature set is additionally extended to include customer prizes.

Initially, each input vector $x_i$ is mapped to a 128-dimensional node embedding $h_i$ via a linear transformation, using distinct parameters for the depot and customer nodes. The embeddings $h_0, \ldots, h_N$ are then processed through several layers. First, two attention layers encode static instance information. Next, the current solution $s$ is integrated via a *message passing layer*, followed by a *tour encoding layer*. The message passing layer facilitates information exchange between consecutive nodes in the solution, while the tour encoding layer computes embeddings for each tour. These two layers are explained in more detail below. Finally, two additional attention layers refine the representations. The attention mechanisms employed are consistent with those used in prior work (e.g., Kwon et al. (2020)), and detailed descriptions are omitted here for brevity.

**Message Passing Layer** The message passing layer updates the embedding of a customer $c_i$ by incorporating information from its immediate neighbors (i.e., nodes that are visited before and after $c_i$ in the solution $s$). Specifically, the embedding $h_i$ of customer $c_i$ is updated as follows:

$$h_i' = \text{Norm}\left(h_i + \text{FF}\left(\text{ReLU}\left(W^3\left[h_i; W^1 h_{\text{prev}(i)} + W^2 h_{\text{next}(i)}\right]\right)\right)\right)$$

In this equation, $\text{prev}(i)$ and $\text{next}(i)$ represent the indices of the nodes immediately preceding and following $c_i$ in the solution $s$. The weight matrices $W^1$ and $W^2$ are used to transform the embeddings of these neighboring nodes, while $W^3$ is applied to the concatenated vector of $h_i$ and the aggregated embeddings from the neighbors. The ReLU activation function introduces non-linearity into the transformation. The output of this transformation is processed through a feed-forward network, which consists of two linear layers with a ReLU activation function in between. The resulting output, combined with the original embedding $h_i$ via a skip connection, is then normalized using instance normalization.

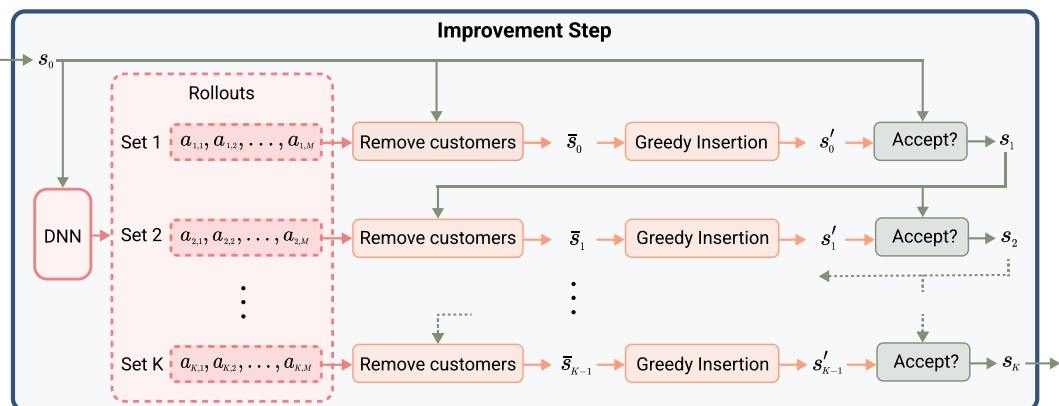

Figure 2: Improvement step of NDS.

**Tour Encoding Layer** The tour encoding layer updates the embedding of each customer $c_i$ by incorporating information from the tour they are part of. To this end, a tour embedding is first computed using mean aggregation of the embeddings of all customers within the same tour, and this aggregated tour embedding is then used to update the individual customer embeddings. Specifically, the embedding $h_i$ of customer $c_i$ is updated as follows:

$$\hat{h}_i = \text{Norm}\big(h'_i + \text{FF}\big(\text{ReLU}\big(W^4\big[h'_i; \sum\nolimits_{j \in \mathcal{T}(i)} h'_j\big]\big)\big)\big),$$

where $\mathcal{T}(i)$ denotes the set of customers in the same tour as customer $c_i$ and $W^4$ is a weight matrix. This layer captures important information about which customers belong to the same tour in the current solution, without considering their specific positions within the tour.

### 4.3.2 Decoder

Given the node embeddings generated by the encoder, the decoder is responsible for sequentially selecting customers for removal. The overall architecture of our decoder is identical to that of Hottung et al. (2025), which utilizes a multi-head attention mechanism (Vaswani et al., 2017) followed by a pointer mechanism (Vinyals et al., 2015). This architecture has been widely used in many routing problem methods (Kool et al., 2019; Kwon et al., 2020).

Our approach differs from previous works in that we account for the already selected customers at each decision step. This contrasts with construction-based methods, where each decision is independent of prior selections. To achieve this, we integrate a gated recurrent unit (GRU) (Cho, 2014), which is used to compute the query for the multi-head attention mechanism. At each decision step, the GRU takes the embedding of the previously selected customer as input, updating its internal state to incorporate past decisions.

### 4.4 Search

At test time, we leverage the learned policy within a search framework that supports batched rollouts, enabling fast execution. Importantly, this framework is problem-agnostic, meaning it contains no problem-specific components, allowing it to be applied to a broader range of problems than those evaluated in this paper.

Our search framework consists of two main components: the improvement step function (illustrated in Figure 2) and the high-level augmented simulated annealing (ASA) algorithm (Algorithm 2). The improvement step function aims to enhance a given solution by iteratively applying the policy model through a series of deconstruction and reconstruction steps. The ASA algorithm integrates this function and supports batched execution for improved performance on the GPU. It is important to note that we parallelize solely on the GPU, requiring only a single CPU core during test time.

---

**Algorithm 2** Augmented Simulated Annealing

---

1: **procedure** SEARCH(Instance $l$, Number of iterations $maxIter$, number of augmentations $A$, number of rollouts $K$, start temperature $\lambda^{start}$, temperature decay rate $\lambda^{decay}$, trained policy network $\pi_\theta$, threshold factor $\delta$)
2:      $\lambda \leftarrow \lambda^{start}$
3:      $\{l'_1, l'_2, \ldots, l'_A\} \leftarrow$ CREATEAUGMENTATIONS($l$)
4:      $s_a \leftarrow$ GENERATESTARTSOLUTION($l'_a$)     $\forall a \in \{1, \ldots, A\}$
5:      **for** $iter = 1, \ldots, maxIter$ **do**
6:         $s_a \leftarrow$ IMPROVEMENTSTEP($s_a, \pi_\theta, \lambda, K$)     $\forall a \in \{1, \ldots, A\}$
7:         $cost_a \leftarrow$ OBJ($s_a$)     $\forall a \in \{1, \ldots, A\}$
8:         $cost^* \leftarrow \min(cost_0, \ldots, cost_A)$
9:         $thresh \leftarrow cost^* + (\lambda \times \delta)$
10:        **for** $a = 1, \ldots, A$ **do**
11:           **if** $cost_a > thresh$ **then**
12:             $s_a \leftarrow$ RANDOMCHOICE($\{s' \in \{s_0, \ldots s_A\} \mid$ OBJ($s'$) $< thresh\}$)
13:           **end if**
14:        **end for**
15:        $\lambda \leftarrow$ REDUCETEMPERATURE($\lambda, \lambda^{decay}$)
16:      **end for**
17: **end procedure**

---

**Improvement Step**   The improvement step, the core component of the overall search algorithm, is depicted in Figure 2. Note that the improvement step procedure is designed to limit the data transfer between GPU and CPU while exploring a large number of different solutions. The process begins with an initial solution $s_0$ that is passed to the policy DNN, which generates $K$ rollouts, each consisting of $M$ actions that specify the customers to be removed. Once the policy DNN completes its execution, these rollouts are sequentially applied to produce new candidate solutions. Specifically, the solution $s_0$ is first deconstructed based on the actions from the first rollout (yielding $\bar{s}_0$) and then reconstructed into $s'_0$. After reconstruction, a simulated annealing (SA) based acceptance criterion is used to determine whether $s'_0$ or $s_0$ should be retained, resulting in $s_1$. This process is repeated in each subsequent iteration. After $K$ iterations, the final solution $s_K$ is returned, representing the outcome of $K$ consecutive deconstruction and reconstruction operations. By performing these iterations sequentially, the solution $s_0$ is significantly modified, often leading to notable cost improvements between the initial input $s_0$ and the final output $s_K$, while only needing to transfer data from the GPU to the CPU a single time.

**Augmented Simulated Annealing**   We introduced a novel simulated annealing (SA) algorithm to conduct a high-level search specifically designed for GPU-based parallelization. While parallel SA algorithms have been proposed in prior work, (Ferreiro et al., 2013; Jeong & Kim, 1990; Onbaşoğlu & Özdamar, 2001), their main concern is on the information exchange between CPU cores. In contrast, our approach focuses on executing parallel rollouts of the policy network on the GPU.

At a high level, the ASA technique, shown in Algorithm 2, modifies solutions over multiple iterations using a temperature-based acceptance criterion. This criterion allows worsening solutions to be accepted with a certain probability, which depends on the current temperature. The temperature $\lambda$ is manually set at the start of the search (line 2) and is gradually reduced after each iteration (line 15), resulting in a decreasing probability of accepting worsening solutions during the improvement step (line 6). For a detailed discussion on SA, we refer the reader to Gendreau et al. (2010).

To enable parallel search for a single instance, we employ the augmentation technique introduced in Kwon et al. (2020), which creates a set of augmentations $l'_1, l'_2, \ldots, l'_A$ for an instance $l$. The search is then conducted in parallel for these augmentations. After each modification by the improvement step procedure (line 6), solutions can be exchanged between different augmentations. Specifically, we iterate over all augmentation instances (lines 10 to 14) and replace solutions that surpass a certain cost threshold with randomly selected solutions whose costs fall below the threshold. This threshold is calculated based on the cost of the current best solution and the temperature, adjusted by a factor $\delta > 1$, as shown in line 9. The goal is to replace solutions that are unlikely to surpass the quality of the current best solution, given the current temperature.

# 5    Experiments

We evaluate NDS on three VRP variants with 100 to 2000 customers and compare to state-of-the-art learning-based and traditional OR methods. Additionally, we provide ablation experiments for the individual components of NDS and evaluate the generalization across different instance distributions. All experiments are conducted on a research cluster utilizing a single Nvidia A100 GPU per run. Our implementation of NDS is available at `https://github.com/ahottung/NDS`.

## 5.1    Problem Instances

**CVRP**    We use the instance generator from Kool et al. (2019) to create scenarios with uniformly distributed customer locations, and the generator from Queiroga et al. (2022) for generating more realistic instances with clustered customer locations to better simulate real-world conditions.

**VRPTW**    We use the instance generator from Queiroga et al. (2022) to generate customer locations and demands, while time windows are generated following the methodology outlined by Solomon (1987).

**PCVRP**    To generate PCVRP instances, we use the instance generator from Queiroga et al. (2022) to create customer locations and demands. Customer prize values are generated at random, with higher prizes assigned to customers with greater demand, reflecting the increased resources required to service them.

## 5.2    Search Performance

**Baselines**    We compare NDS to several heuristic solvers, including HGS (Vidal, 2022), SISRs (Christiaens & Vanden Berghe, 2020), and LKH3 (Helsgaun, 2017). Additionally, we include PyVRP (Wouda et al., 2024) (version 0.9.0), which is an open-source extension of HGS for other VRP variants. For the CVRP, we further compare NDS to the state-of-the-art learning-based methods, SGBS-EAS (Choo et al., 2022), NeuOpt (Ma et al., 2023), BQ (Drakulic et al., 2023), LEHD (Luo et al., 2023), UDC (Zheng et al., 2024), and GLOP (Ye et al., 2024b). We run all baselines ourselves on the same test instances. More details on the baseline settings can be found in Appendix A.

**NDS Training**    For each problem and problem size, we perform a separate training run. Training consists of 2000 epochs for settings with 1000 or fewer customers. For the 2000 customer setting, we resume training from the 1000 customer model checkpoint at 1500 epochs and train for an additional 500 epochs. In each epoch, we process 1500 instances, with each instance undergoing 100 iterations, 128 rollouts, and 10 initial improvement steps. The learning rate is set to $10^{-4}$ and 15 customers are selected per deconstruction step across all problem sizes. The training durations are approximately 5, 8, 15, and 8 days for the problem sizes 100, 500, 1000 and 2000, respectively. The training curves are presented in Appendix B, while visualizations of policy rollouts are available in Appendix C.

**Evaluation Setup**    At test time, we limit the runtime to 5, 60, 120, and 240 seconds of wall time per instance for HGS, SISRs, and NDS to ensure a fair comparison, as these methods process test instances sequentially. For most learning-based approaches we report results for two different termination criteria, i.e., the original termination criteria proposed in the respective paper and a new wall-time-based termination criteria. For approaches that process instances in batches, an adjusted time limit is applied to each batch as a whole, ensuring that the total time required to solve the complete set matches that of approaches using a per-instance time limit. Note that this experimental setup favors batch-based approaches over NDS, as processing instances in batches is computationally more efficient, but not feasible in all applications. All approaches are restricted to using a single CPU core. For the CVRP, we use the test instances from Kool et al. (2019) for $N{=}100$ (10,000 instances), Drakulic et al. (2023) for $N{=}500$ (128 instances), and Ye et al. (2024b) for $N{=}1000$ and $N{=}2000$ (100 instances each). For the VRPTW and PCVRP, we generate new test sets consisting of 10,000 instances for $N{=}100$ and 250 instances for settings with more than 100 customers. To allow for a fair comparison, we ensure that all reinforcement learning based approaches are trained on instances from the same distribution that is also used for sampling the test instances.

Table 1: Performance on test data. The gap is calculated relative to HGS for the CVRP and relative to PyVRP-HGS for the VRPTW and PCVRP. **Runtime is reported on a per-instance basis** in seconds. The best results (i.e., those with the lowest objective function value) are shown in bold, and the second-best are underlined.

| Method | | N=100 | | | N=500 | | | N=1000 | | | N=2000 | | |
|---|---|---|---|---|---|---|---|---|---|---|---|---|---|
| | | Obj. | Gap | Time | Obj. | Gap | Time | Obj. | Gap | Time | Obj. | Gap | Time |
| **CVRP** | HGS | **15.57** | - | 5 | 36.66 | - | 60 | 41.51 | - | 121 | 57.38 | - | 241 |
| | SISRs | 15.62 | 0.32% | 5 | 36.65 | 0.01% | 60 | 41.14 | -0.83% | 120 | 56.04 | -2.27% | 240 |
| | LKH3 | 15.64 | 0.50% | 41 | 37.25 | 1.66% | 174 | 42.16 | 1.61% | 408 | 58.12 | 1.35% | 1448 |
| | SGBS-EAS   Iter. Limit | 15.59 | 0.20% | 2 | - | - | - | - | - | - | - | - | - |
| | Time Limit | 15.59 | 0.17% | 5 | - | - | - | - | - | - | - | - | - |
| | NeuOpt   10k Iter. | 15.66 | 0.59% | 1 | - | - | - | - | - | - | - | - | - |
| | Time Limit | 15.59 | 0.15% | 5 | - | - | - | - | - | - | - | - | - |
| | BQ   Beam width 16 | 15.81 | 1.55% | 1 | 37.64 | 2.68% | 10 | 43.53 | 4.86% | 45 | 61.75 | 7.61% | 323 |
| | Beam width 64 | 15.74 | 1.13% | 1 | 37.51 | 2.32% | 23 | 43.32 | 4.36% | 164 | - | - | - |
| | LEHD   1000 Iter. | 15.63 | 0.41% | 1 | 37.10 | 1.21% | 28 | 42.41 | 2.17% | 159 | 59.45 | 3.60% | 1476 |
| | Time Limit | 15.61 | 0.30% | 5 | 37.04 | 1.04% | 60 | 42.47 | 2.31% | 121 | 60.11 | 4.76% | 246 |
| | UDC   250 Stages | - | - | - | 37.71 | 2.86% | 13 | 42.77 | 3.04% | 21 | - | - | - |
| | Time Limit | - | - | - | 37.63 | 2.67% | 60 | 42.65 | 2.76% | 121 | - | - | - |
| | GLOP (LKH3) | - | - | - | - | - | - | 45.90 | 10.58% | 1 | 63.02 | 9.82% | 2 |
| | NDS | 15.57 | 0.04% | 5 | **36.57** | -0.20% | 60 | **41.11** | -0.90% | 120 | **56.00** | -2.34% | 240 |
| **VRPTW** | PyVRP-HGS | 12.98 | - | 5 | 49.01 | - | 60 | 90.35 | - | 120 | 173.46 | - | 240 |
| | SISRs | 13.00 | 0.20% | 5 | 48.09 | -1.87% | 60 | 87.68 | -2.98% | 120 | 167.49 | -3.49% | 240 |
| | SGBS-EAS   Default | 13.15 | 1.35% | 3 | - | - | - | - | - | - | - | - | - |
| | Time Limit | 13.13 | 1.22% | 5 | - | - | - | - | - | - | - | - | - |
| | NDS | **12.95** | -0.19% | 5 | **47.94** | -2.17% | 60 | **87.54** | -3.14% | 120 | **167.48** | -3.50% | 240 |
| **PCVRP** | PyVRP-HGS | 10.11 | - | 5 | 44.97 | - | 60 | 84.91 | - | 120 | 165.56 | - | 240 |
| | SISRs | 9.94 | -1.66% | 5 | 43.22 | -3.90% | 60 | 81.12 | -4.55% | 120 | 158.17 | -4.54% | 240 |
| | NDS | **9.90** | -2.07% | 5 | **43.12** | -4.12% | 60 | **80.99** | -4.71% | 121 | **158.09** | -4.60% | 241 |

**NDS Test Configuration**   For NDS, the starting temperature $\lambda^{start}$ is set to 0.1 and decays exponentially to 0.001 throughout the search. The threshold factor $\delta$ is fixed at 15. During the improvement step, 200 rollouts are performed per instance, and each deconstructed solution is reconstructed 5 times ($1\times$ based on the selected order of the DNN and $4\times$ using a random customer order). The number of augmentations is set to 8 for the CVRP and VRPTW, and 128 for the PCVRP. Note that this configuration was manually selected rather than derived from automated hyperparameter tuning. In Appendix E, we analyze the impact of the hyperparameters on search performance.

**Results**   Table 1 presents the performance of all compared methods on the test data. The gap is reported relative to HGS for the CVRP, and to PyVRP-HGS for the VRPTW and PCVRP. By choosing HGS for the gap calculation, we follow recent publications and ensure easy comparability of our results with these earlier works. Across the 12 test settings, NDS delivers the best performance in 11 cases, with HGS being the only approach able to outperform it on the CVRP with 100 customers. Compared to other learning-based methods, NDS shows significant performance improvements across all CVRP sizes. On the CVRP with 2000 customers, NDS achieves a 6.83% improvement over the best-performing learning-based method, LEHD (when both are given the same runtime), and an 11.13% improvement over GLOP. Against the state-of-the-art HGS and its extension, PyVRP-HGS, NDS performs especially well on larger instances, achieving an improvement of more than 2% across all problems for instances with 2000 customers. For the PCVRP, NDS also attains substantial improvements relative to PyVRP-HGS, exceeding 4% on instances with 500 or more

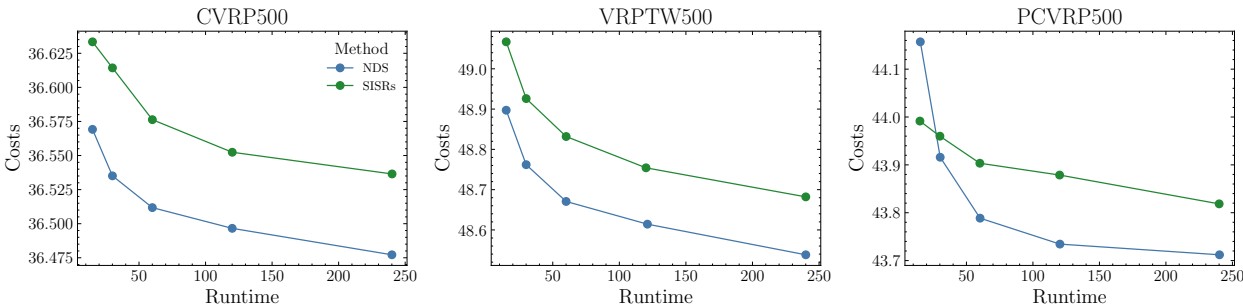

Figure 3: Comparison of NDS and SISRs performance across varying runtime limits.

nodes. When compared to SISRs, NDS maintains a small advantage on larger instances and demonstrates significantly better performance on small instances.

## 5.3 Performance Across Different Runtime Limits

To better understand the application scenarios of NDS, we evaluate its performance under varying runtime limits. In this experiment, we compare NDS only to SISRs as it is the overall best performing baseline in the previous experiment (see Section 5.2). We use the same trained models and evaluation setup as in the previous experiment, but with test sets reduced to 100 instances each to manage computational costs.

Figure 3 presents the results for instances with 500 customers. For each problem, five different runtime limits ranging from 15 to 250 seconds are considered. The hyperparameters of SISRs and NDS remain consistent with those used in previous experiments across all runtime limits. Notably, NDS outperforms SISRs across all runtime limits, except for the PCVRP, where SISRs perform better under a 15-second runtime limit. The relative performance gap is particularly striking for the CVRP, where running NDS for just 30 seconds surpasses the performance of SISRs at 250 seconds. Similar trends are observed for instances with 100, 1000, and 2000 customers. Additional details and results for other problem sizes are provided in Appendix D.

## 5.4 Ablation Studies

We perform a series of ablation experiments to assess the importance of different components of NDS. These experiments are conducted on separate validation instances with $N{=}500$ customers. The parameter configuration remains identical to the previous section, except the training is reduced to 1,000 epochs and the ASA search is limited by the number of iterations. For the CVRP and VRPTW, we run 1,000 iterations using 8 augmentations, while for the PCVRP, we perform 50 iterations with 128 augmentations. In addition to the ablation results reported here, we present an ablation study on the impact of the initial improvement step during training in Appendix F.

**Network Architecture** We assess the impact of the message passing layer (MPL) and tour encoding layer (TEL) on overall performance by training separate models without these components. Table 2a summarizes the resulting search performance. Excluding both layers leads to a significant performance drop, with a 1.5% reduction on the PCVRP. Even the removal of a single layer causes a notable performance decline, particularly for the VRPTW and PCVRP. The VRPTW in particular benefits from both layers, likely due to the MPL's ability to better interpret and handle time windows.

**Insertion Order** The insertion algorithm reinserts removed customers in a specified order. During testing, we reconstruct a deconstructed solution five times using different customer orders and retain the best solution. For the first reconstruction iteration, we use the customer order provided by the DNN, while for the remaining four iterations we use a random order. We compare this standard setting to using only random orders across all five insertion iterations to assess whether the DNN has learned to select an order that improves the overall search performance. The results in Table 2b show that using a only random orderings leads to significantly

Table 2: Ablation experiments. All tables report objective function values.

(a) Impact of the message passing layer (MPL) and the tour encoding layer (TEL) on performance.

| MPL | TEL | CVRP | VRPTW | PCVRP |
|-----|-----|------|-------|-------|
| ✓ | ✓ | **36.81** | **47.68** | **42.96** |
| ✓ | ✗ | 36.82 | 47.75 | 43.13 |
| ✗ | ✓ | 36.81 | 47.74 | 42.98 |
| ✗ | ✗ | 36.87 | 47.87 | 43.62 |

(b) Insertion order

| Order | CVRP | VRPTW | PCVRP |
|-------|------|-------|-------|
| DNN+Random | **36.81** | **47.68** | **42.96** |
| Random | 36.86 | 47.76 | 43.05 |

(c) Deconstruction policy

| Policy | CVRP | VRPTW | PCVRP |
|--------|------|-------|-------|
| DNN | **36.81** | **47.68** | **42.96** |
| Heuristic | 37.03 | 48.16 | 43.61 |

Table 3: Out-of-distribution (OOD) vs. in-distribution (ID) performance on the CVRP500.

| Method | Uniform Locations | | | | | | Clustered Locations | | | | | |
| | Low Capacity | | | High Capacity | | | Low Capacity | | | High Capacity | | |
| | Obj. | Gap | Time | Obj. | Gap | Time | Obj. | Gap | Time | Obj. | Gap | Time |
|--------|------|-----|------|------|-----|------|------|-----|------|------|-----|------|
| HGS | 91.73 | - | 60 | 47.89 | - | 60 | 88.20 | - | 60 | 44.53 | - | 61 |
| SISRs | 91.34 | -0.38% | 60 | 47.79 | -0.17% | 60 | 87.78 | -0.48% | 60 | 44.31 | -0.49% | 60 |
| NDS (OOD) | 91.15 | -0.59% | 60 | 47.70 | -0.36% | 60 | 87.75 | -0.53% | 60 | 44.29 | -0.54% | 60 |
| NDS (ID) | 91.14 | -0.59% | 60 | 47.69 | -0.38% | 60 | 87.70 | -0.58% | 60 | 44.26 | -0.60% | 60 |

worse performance across all three problems, indicating that the learned policy not only plays a crucial role in deconstruction, but also significantly influences reconstruction.

**Learned Policy**   We assess the relevance and effectiveness of the learned deconstruction policy by replacing it with a handcrafted heuristic based on the destroy procedure outlined in Christiaens & Vanden Berghe (2020). The resulting approach eliminates any learned components, but is otherwise identical to NDS. The performance comparison, shown in Figure 2c, reveals that the heuristic deconstruction policy performs significantly worse than the learned counterpart, with performance gaps of up to 1.5% on the PCVRP. This demonstrates that the DNN is capable of learning a highly efficient policy that surpasses handcrafted methods.

## 5.5   Generalization

One major advantage of learning-based solution approaches is their ability to adapt precisely to the specific type of instances at hand. However, in real-world scenarios, concept drift in the instance distributions cannot always be avoided. In this experiment, we evaluate whether the learned policies of NDS can handle instances sampled from a slightly different distribution. For the CVRP with $N$=500, we train a policy on instances with medium-capacity vehicles and customer locations that follow a mix of uniform and clustered distributions. We then evaluated the learned policy on instances with low- and high-capacity vehicles, and customer locations following either uniform or clustered distributions. Additionally, we train distribution-specific models for each test setting for comparison. As a baseline, we compare against HGS and SISRs, giving all approaches the same runtime. The results are shown in Table 3, where NDS (OOD) represents the model's performance when the training and test distributions differ, and NDS (ID) represents the setting where the training and test distributions are identical. Overall, the performance difference between the two settings is minimal, indicating that NDS generalizes well across different distributions. Interestingly, the distribution of customer locations has a larger impact on performance than vehicle capacity. In Appendix G, we also evaluate NDS's ability to generalize to both smaller and significantly larger instances.

### 5.6 Scalability Analysis

We assess the scalability of NDS by analyzing its runtime and GPU memory consumption on CVRP instances of varying sizes. Figure 4 presents the relative resource usage as a function of problem size. Overall, NDS demonstrates strong scalability to larger instances. Notably, solving instances with 1,000 customers requires only 61% more runtime and 23% more memory compared to instances with 100 customers, despite the problem size increasing by an order of magnitude.

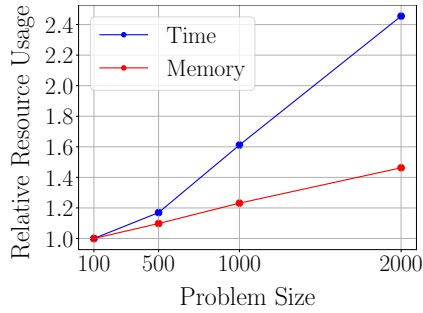

Figure 4: Scalability

## 6 Conclusion

In this work, we introduced a novel search method, NDS, which leverages a learned policy to deconstruct solutions for routing problems. NDS presents several key advantages. First, it delivers excellent performance, consistently matching or outperforming state-of-the-art OR methods under equal runtime. Second, NDS scales effectively to larger problem instances, handling up to $N=2000$ customers, due to the fact that the number of customers selected by the policy is independent of the problem size. Third, it demonstrates strong generalization across different data distributions. Finally, NDS is easily adaptable to new vehicle routing problems, requiring only small adjustments to the greedy insertion heuristic and the model input.

A notable limitation is the reliance on a GPU for executing the policy network. Future research could explore model distillation techniques to lower the computational requirements or investigate whether the underlying principles of the learned policies can be approximated using faster, more efficient algorithms. Another limitation of NDS is its reliance on training data that closely resembles the instances encountered during testing. Future work could investigate whether fine-tuning techniques enable fast adaptation of a pretrained foundation model to new problem instances. This could significantly reduce training time and minimize the amount of required training data.

### Acknowledgments

André Hottung was supported by the Deutsche Forschungsgemeinschaft (DFG, German Research Foundation) under Grant No. 521243122. Paula Wong-Chung received funding from the Deutscher Akademischer Austauschdienst (DAAD) RISE Germany program for this research. Additionally, we gratefully acknowledge the funding of this project by computing time provided by the Paderborn Center for Parallel Computing (PC2). Furthermore, some computational experiments in this work have been performed using the Bielefeld GPU Cluster. We thank the HPC.NRW team for their support.

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

# A    Baseline Configurations

**HGS**    We conduct evaluation runs using the publicly available[1] implementation from the authors that has been designed and calibrated for medium-scale CVRP instances with up to 1,000 customers. We use a wall-time-based stopping criteria to limit the duration of each search run.

**SISRs**    Since the code of SISRs is not publicly available, we reimplement SISRs in C++ based on the description of the authors. Our implementation matches the performance reported in the original paper. We use a time based stopping criteria to limit the duration of each search run.

**SGBS-EAS**    We conduct evaluation runs using the code that was made publicly available[2] by the authors. For the CVRP with 100 customers, we use the publicly available model which has been trained on instances from the same distribution as our test instances. We run SGBS-EAS using the configuration proposed in the original paper in which search runs are limited based on the number of iterations. Additionally, we conduct experiments in which we limit the search to the same runtime as NDS. More precisely, we limit the total runtime to 13.9 hours, which corresponds to 5 seconds per instance. We increase the beam width to 7 in this experiment to increase exploration and to better use the available runtime. For the VRPTW with 100 customers, no pretrained models are publicly available. We hence train the models ourselves on instances sampled from the same distribution as our test instances to allow for fair comparison. We use the same training configuration as in the original paper (Kwon et al., 2020) and train the model for 10,000 epochs with each epoch comprising 10,000 instances. At test time, we use the same setup for the VRPTW as for the CVRP.

**NeuOpt**    We evaluate NeuOpt on the CVRP with 100 customers using the code and trained model made available[3] by the authors. Note that the model has been trained on CVRP instances sampled from the same distribution as the test instances. Like SGBS-EAS, NeuOpt is evaluated only on instances with 100 customers, as its autoregressive nature makes training on larger instances impractical. We conduct two sets of experiments: one where the search is limited to 10,000 iterations (as in the original paper) and another where the search is constrained by runtime, using the same runtime limits applied to NDS. For the former, we set the dynamic data augmentation (D2A) parameter to 1, and for the latter, we set D2A to 5 to enhance exploration during the search.

**BQ**    We conduct evaluation runs using the code and trained model made publicly available[4] by the authors. BQ is a supervised learning approach that focuses on generalization and requires near-optimal solutions during training, which are only obtainable for smaller instances. Consequently, we use the publicly available model, trained on instances with 100 customers, to solve all four problem sizes considered in our experiments. Note that the model has been trained on the same distribution as our test instances with 100 customers. We perform two sets of experiments: one with a beam width of 16 and another with a beam width of 64. In both sets, we use a batch size of 1, as the beam search constructs multiple solutions in parallel, fully utilizing the parallel computing capabilities of our GPUs. For the CVRP with 2000 customers, results are only reported for a beam width of 16, as a beam width of 64 leads to out-of-memory issues.

**LEHD**    We evaluate LEHD using the publicly available code[5] provided by the authors. LEHD is a supervised learning approach focused on generalization and requires near-optimal solutions during training, which are only obtainable for smaller instances. Consequently, we use the publicly available model, trained on instances with 100 customers, to solve all four problem sizes considered in our experiments. Note that the model was trained on the same distribution as our test instances with 100 customers. We evaluate LEHD using the random re-construct method proposed by the authors. We conduct two sets of experiments: one where the search is limited to 1,000 iterations (as in the original paper) and another where the search is constrained by runtime, using the same runtime limits applied to NDS.

---

[1]`https://github.com/vidalt/HGS-CVRP`
[2]`https://github.com/yd-kwon/SGBS`
[3]`https://github.com/yining043/NeuOpt`
[4]`https://github.com/naver/bq-nco`
[5]`https://github.com/CIAM-Group/NCO_code/tree/main/single_objective/LEHD`

**UDC**  We evaluate UDC using the code that was made publicly available[6] by the authors. For a fair comparison, we use custom models that are trained specifically for each considered problem size, instead of using the provided one-size-fits-all trained model. We use the default training configuration, but change `sample_size` to 30 for the CVRP with 1,000 customers to reduce GPU memory usage. For the CVRP with 2,000 customers, we are not able to conduct training due to memory constraints. The training instances are sampled from the same distribution as our test instances.

We evaluate UDC's test-time performance in two sets of experiments: one with the search limited to 250 stages (as in the original paper) and another constrained by runtime, using the same limits applied to NDS. For the CVRP with 500 customers, a batch size of 6 is used, while for the CVRP with 1,000 customers, a batch size of 4 is applied. The batch size is chosen to fully leverage the available GPU memory.

**GLOP**  We evaluate GLOP using the code and trained models provided by the authors[7]. GLOP is a divide-and-conquer approach tailored for large-scale problems, and we evaluate it only on CVRP instances with 1,000 and 2,000 customers, as these are the instance sizes for which trained models are available. The training instances used by the original authors are sampled from the same distribution as our test instances. We run GLOP with the configuration outlined in the original paper, using LKH3 as a subsolver and a single CPU core. Notably, there is no straightforward way to extend GLOP to longer runtime settings for the CVRP.

---

[6] https://github.com/CIAM-Group/NCO_code/tree/main/single_objective/UDC-Large-scale-CO-master
[7] https://github.com/henry-yeh/GLOP

# B    Training Curves

Figure 5 presents the validation performance throughout training for all experiments conducted across the three problem types and four problem sizes. Note that the training of the models for $N$=2000 is warm-started using the model weights from $N$=1000 after 1,500 epochs.

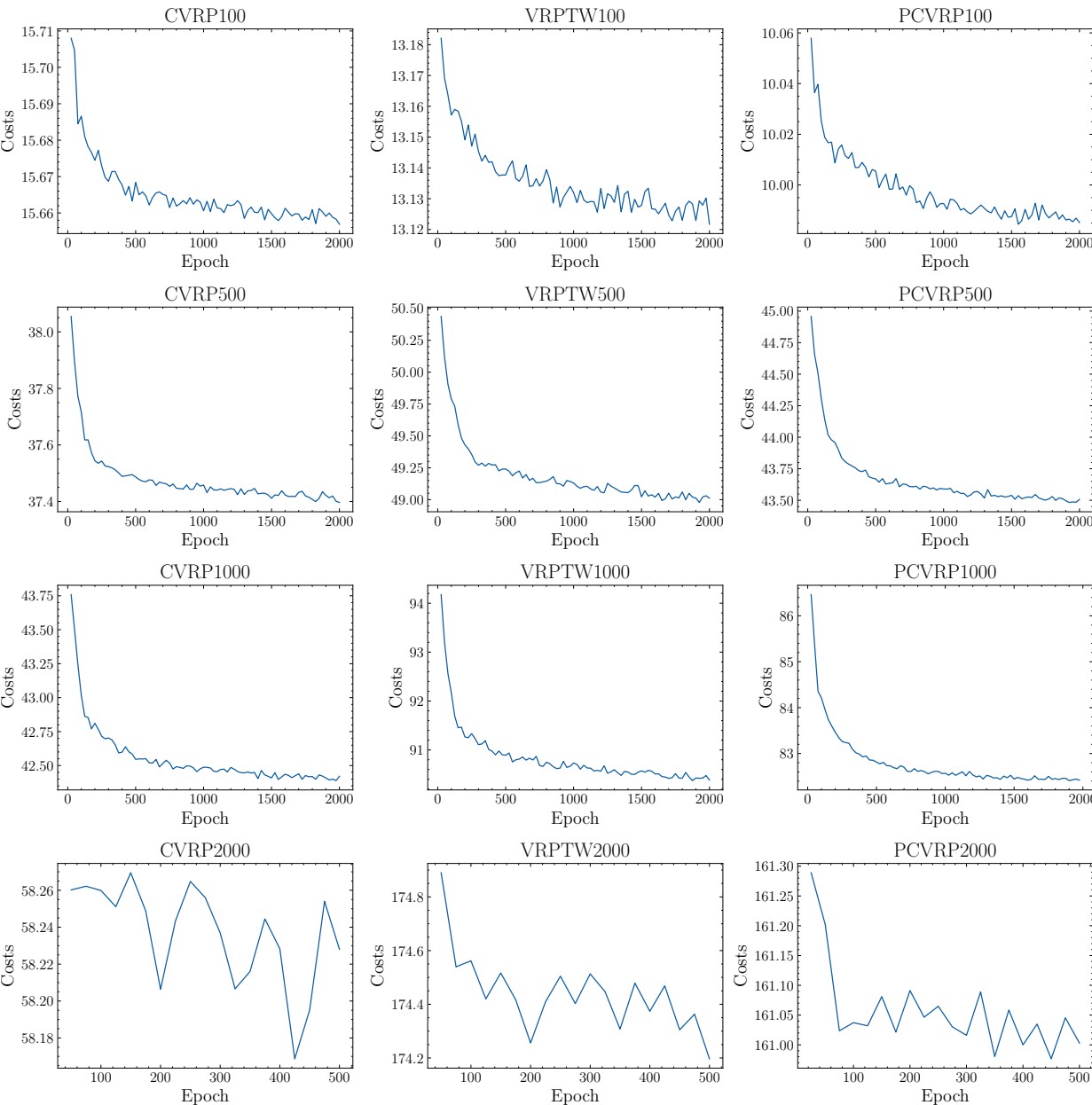

Figure 5: Validation performance throughout the training process.

## C  Visualizations of Policy Rollouts

Figures 6, 8, and 7 show visualizations of different policy rollouts for the CVRP, PCVRP, and VRPTW, respectively. For each problem, we display two different instances, and for each instance, six rollouts are shown. Customers selected for deconstruction are circled in red. We note that the nodes selected for each deconstruction differs, sometimes significantly, allowing NDS to try out a variety of options in each iteration.

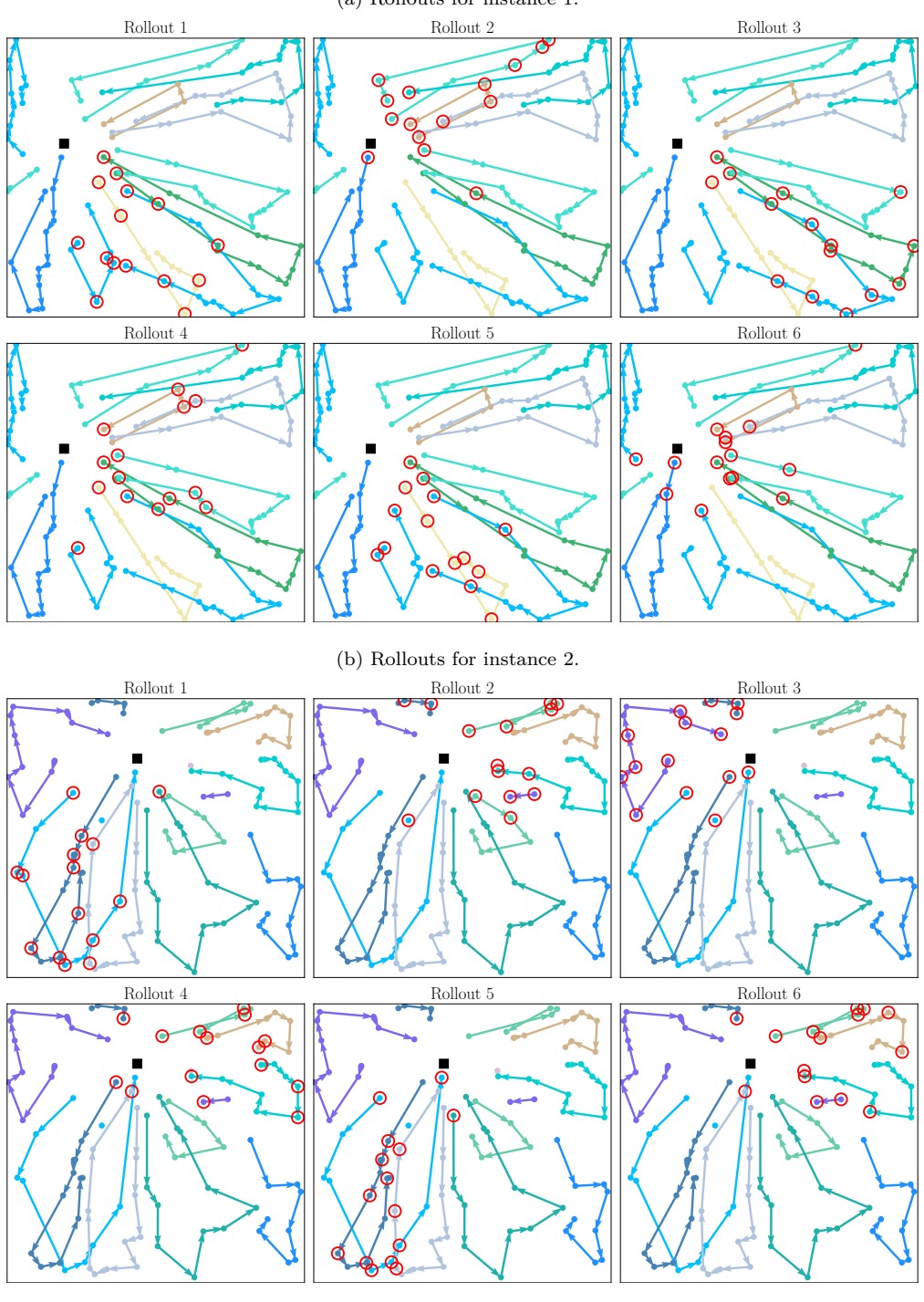

Figure 6: Rollouts for two selected instances for the CVRP with $N$=100 (best viewed in color).

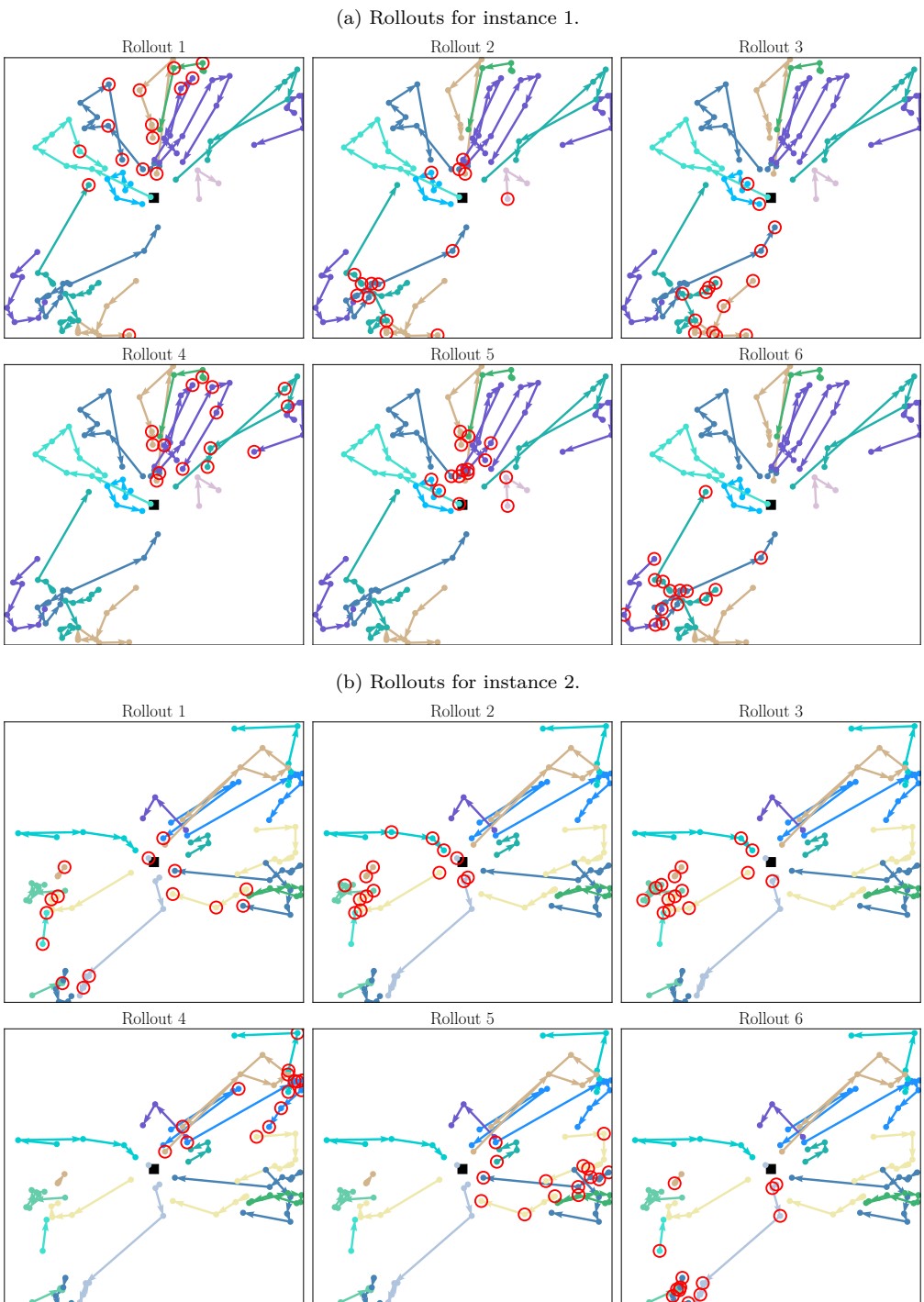

Figure 7: Rollouts for two selected instances for the VRPTW with $N{=}100$ (best viewed in color).

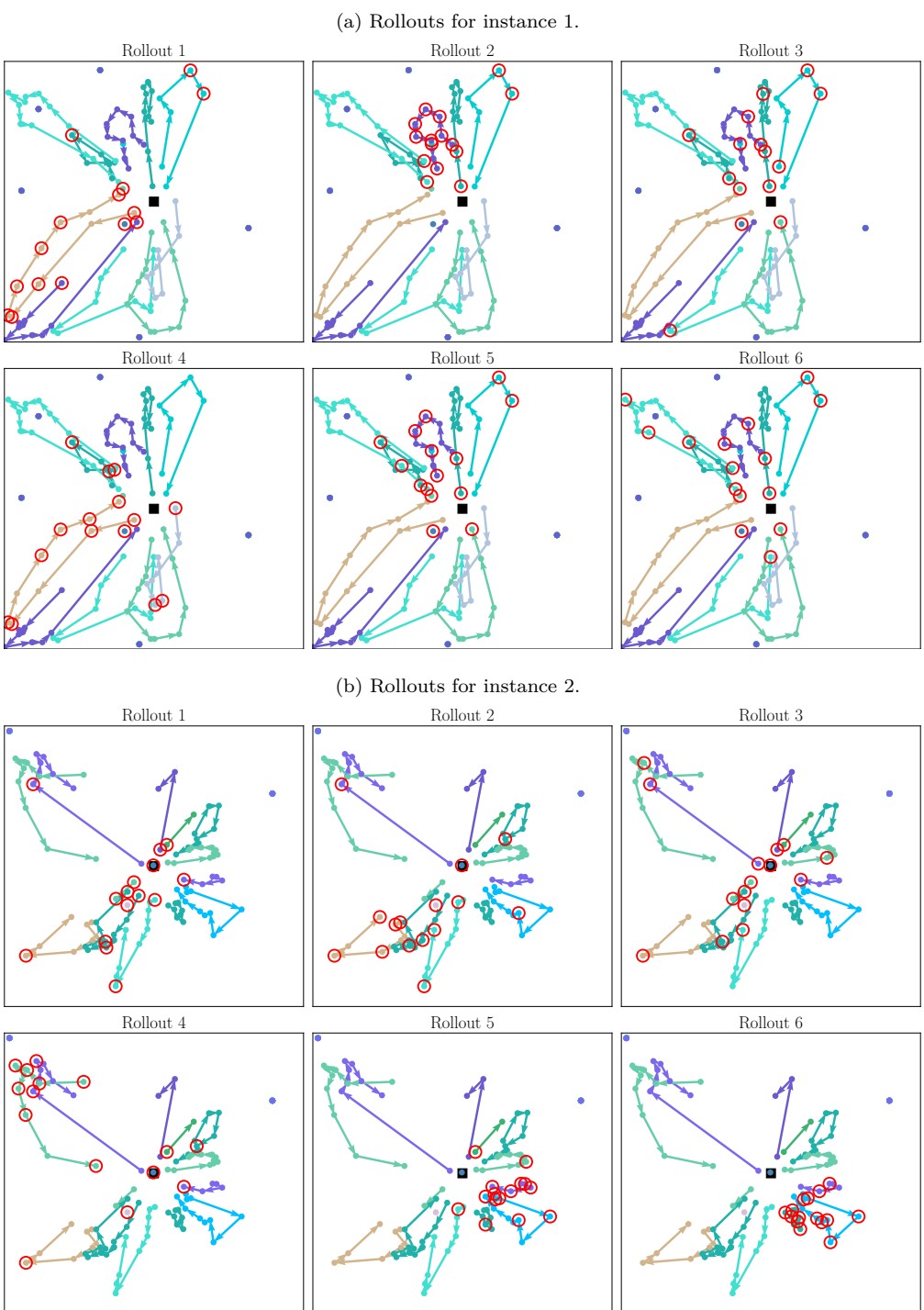

Figure 8: Rollouts for two selected instances for the PCVRP with $N$=100 (best viewed in color).

# D    Performance Across Different Runtime Limits

We evaluate the performance of NDS and SISRs under varying runtime limits. The results are based on the same trained models and evaluation setup described in Section 4.4, but with test sets reduced to 100 instances each to manage computational costs. The hyperparameters remain as described in Section 4.4, except for the number of augmentations for the PCVRP. Specifically, for the PCVRP with 1000 customers, 256 augmentations are used for runtime limits of 240 seconds or more. For the PCVRP with 2000 customers, 64 augmentations are applied for runtime limits of 240 seconds or less, and 256 augmentations for a runtime limit of 16 minutes. The runtime limits in this experiment are selected based on instance size, ranging from 2 seconds for instances with 100 customers to 16 minutes for instances with 2000 customers.

Figure 9 presents the results across all problems and problem sizes. Overall, NDS significantly outperforms SISRs on all problems. SISRs only surpasses NDS in a few cases under very short runtime limits. The performance gap between NDS and SISRs appears to depend more on problem size than on problem type. For example, on instances with 100 customers, SISRs struggles to achieve the solution quality that NDS attains in just 2 seconds, even after 60 seconds of runtime.

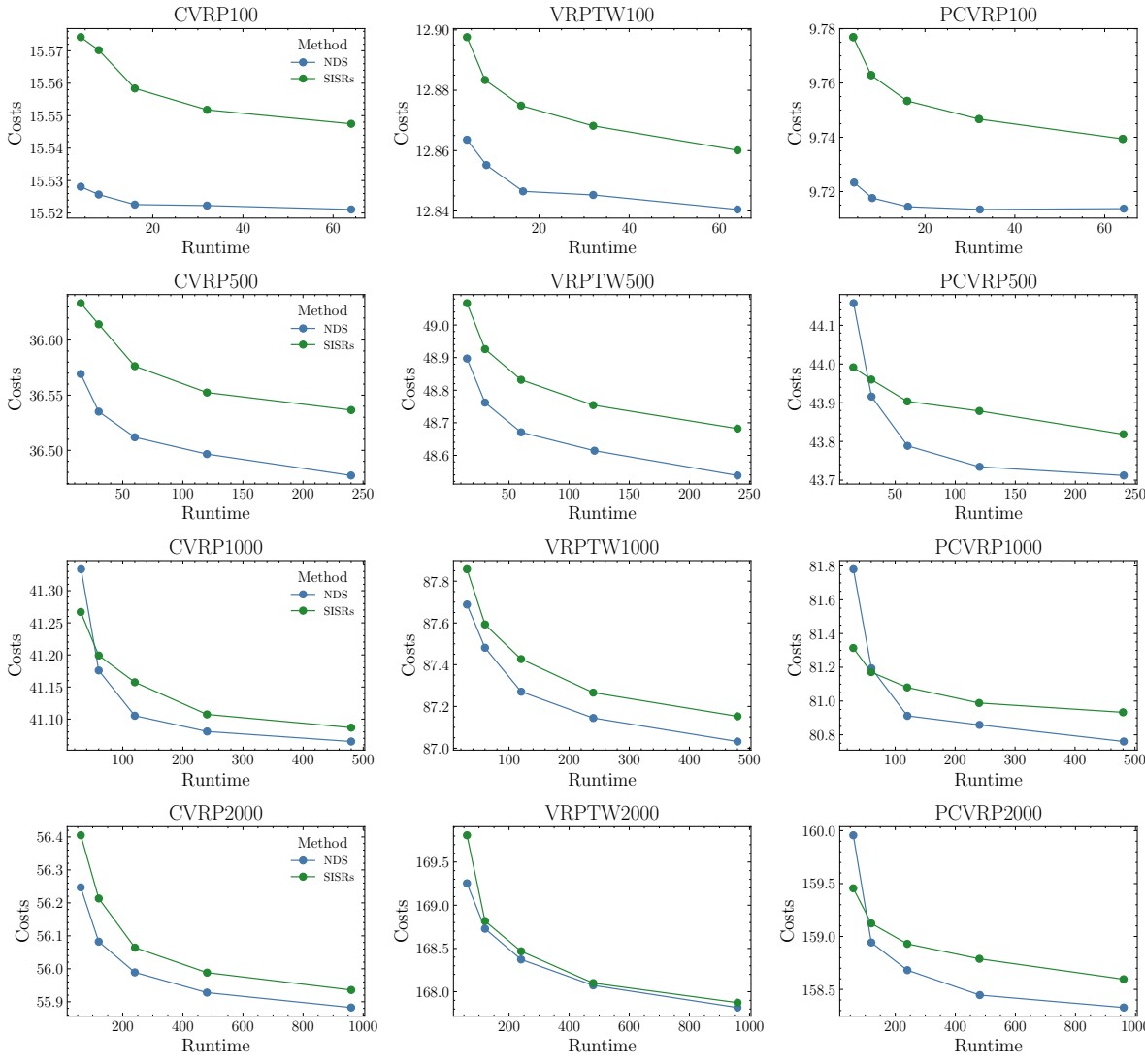

Figure 9: Performance of NDS and SISRs across different runtime limits.

# E    Impact of Hyperparameters on Search Performance

We evaluate the impact of key hyperparameters in the ASA search procedure. To manage computational costs, this experiment is conducted solely on the CVRP with 500 nodes. We perform multiple evaluation runs on the test set, varying the number of augmentations $A$, the number of rollouts $K$, and the threshold factor $\delta$, while keeping all other parameters at their default values as specified in Section 5.2.

The results, presented in Figure 11, indicate that values of $A$ equal or greater than 8 yield the best performance. The number of rollouts $K$ has little impact on performance within the tested range of 100 to 300. For the threshold factor $\delta$, values above 15 yield consistently strong results. Overall, the search procedure demonstrates robustness to hyperparameter choices, suggesting that fine-tuning is not critical for achieving good performance.

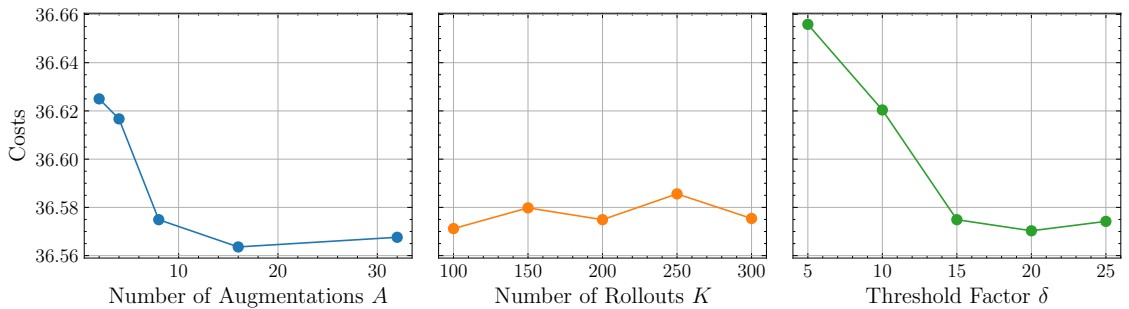

Figure 10: Search performance on the CVRP500 for different hyperparameter setting.

# F    Ablation Study: Initial Improvement Steps

We assess the impact of initial solution improvement through an ablation study. To this end, we train models on problem instances with 500 nodes, omitting any solution improvement steps before the main training phase (lines 10–17 in Algorithm 1). As a result, the model also learns to refine low-quality start solutions.

Figure 11 compares validation performance during training with and without the initial improvement steps. The results reveal substantial performance differences, with training runs that include initial improvement significantly outperforming those without, highlighting the effectiveness of this technique. We hypothesize that initial improvement is beneficial because learning directly from poor-quality start solutions provides limited value; improving them is trivial, whereas most of the search process focuses on refining higher-quality solutions.

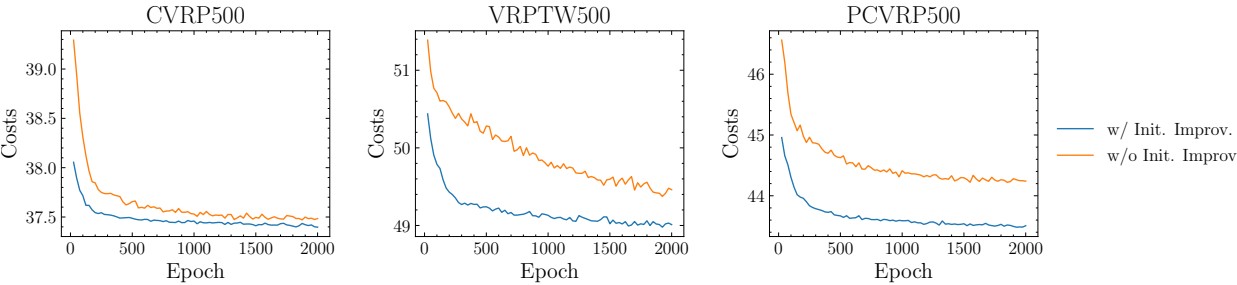

Figure 11: Validation performance for training runs with and without initial solution improvement steps.

# G  Generalization to Smaller and Larger Instances

We also evaluate the generalization ability of NDS to larger and smaller instances by using models trained on instances with 500 customers to solve instances with 100, 1000, and 2000 customers. In this experiment, all hyperparameters are set identically to those in the main experiment.

The results, presented in Table 4, also include in-distribution performance from the main experiment, where instance size-specific models are used, to allow for an easier comparison. The findings indicate that models trained on 500 customer instances perform well on instances with 100 and 1000 customers, with no significant degradation in performance. However, on instances with 2000 customers, a slight drop in performance is observed, particularly for the CVRP.

Table 4: Out-of-distribution (OOD) performance of models trained on instances with 500 nodes, evaluated on both larger and smaller instances, compared to the in-distribution (ID) performance of size-specific models.

| | | N=100 (Smaller) | | | N=1000 (Larger) | | | N=2000 (Much Larger) | | |
|---|---|---|---|---|---|---|---|---|---|---|
| | | Obj. | Gap | Time | Obj. | Gap | Time | Obj. | Gap | Time |
| **CVRP** | HGS | 15.57 | - | 5 | 41.51 | - | 121 | 57.38 | - | 241 |
| | NDS (ID) | 15.57 | 0.04% | 5 | 41.11 | -0.90% | 120 | 56.00 | -2.34% | 240 |
| | NDS (OOD) | 15.59 | 0.13% | 5 | 41.32 | -0.45% | 120 | 57.63 | 0.43% | 240 |
| **VRPTW** | PyVRP-HGS | 12.98 | - | 5 | 90.35 | - | 120 | 173.46 | - | 240 |
| | NDS (ID) | 12.95 | -0.19% | 5 | 87.54 | -3.14% | 120 | 167.48 | -3.50% | 240 |
| | NDS (OOD) | 12.97 | -0.00% | 5 | 87.54 | -3.11% | 120 | 167.76 | -3.28% | 240 |
| **PCVRP** | PyVRP-HGS | 10.11 | - | 5 | 84.91 | - | 120 | 165.56 | - | 240 |
| | NDS (ID) | 9.90 | -2.07% | 5 | 80.99 | -4.71% | 121 | 158.09 | -4.60% | 241 |
| | NDS (OOD) | 9.96 | -1.50% | 5 | 81.05 | -4.55% | 120 | 160.20 | -3.24% | 240 |

