# OpenReview forum: "Neural Deconstruction Search for Vehicle Routing Problems"
_TMLR — Accepted by TMLR_

### Review · Reviewer_LxMb · 2025-02-11

**Summary Of Contributions:**

This paper introduces a deep learning-based simulated annealing approach for solving vehicle routing problems. The motivation and proposed concept are clear and promising: while pure deep learning methods excel in intuitive heuristics but lack precision, traditional OR methods offer accuracy but lack adaptability. The key challenge lies in effectively integrating these two approaches. Experimental results indicate certain advantages of the proposed method. In particular, the out-of-distribution results are impressive. However, while the overall idea is well-articulated, the lack of technical details makes the paper somewhat difficult to follow.

**Audience:**

Yes

**Broader Impact Concerns:**

No broader impact concerns.

**Claims And Evidence:**

Yes

**Requested Changes:**

Methodology descriptions are not clear enough:

1. Section 3.1: The vehicle routing problem is not explicitly defined—what are the input parameters, and what are the expected outcomes? While VRP is a well-known problem, providing a precise definition would improve clarity.

2. Sections 3.1 and 3.3: What constitutes an instance $l$, and how is it encoded in the neural network? The statement that it "involves customers $c_1, \cdots, c_N$" is vague and lacks clarity.

3. Sections 3.1 and 3.3: How is the solution $s$ defined? What features are used to describe a solution in neural networks? This information should be included, at least in the appendix.

Experimental evaluations:

4. Configuration: The paper provides detailed configuration settings but seems not clearly state how these were chosen (correct me if I am wrong). Are all the configurations tuned to the best?

5. Robustness to Configuration Changes: If configuration parameters are perturbed, how does the robustness of the traditional OR method compare to the proposed learning-based method?

**Strengths And Weaknesses:**

Ref to "summary"

---

> ### Author Response · Authors · 2025-02-18
>
> Thank you for your prompt review of our paper and for your valuable comments and suggestions. We truly appreciate your time and insights! Note that we will upload a revised version of the manuscript that includes new experiments and improvements in the coming days.
>
> We are happy that you like the core concept of integrating learning with traditional OR methods and that you find the out-of-distribution results impressive. Please let us address you remaining comments and concerns.
>
> > Section 3.1: The vehicle routing problem is not explicitly defined—what are the input parameters, and what are the expected outcomes? While VRP is a well-known problem, providing a precise definition would improve clarity.
>
> > Methodology descriptions are not clear enough:
> Section 3.1: The vehicle routing problem is not explicitly defined—what are the input parameters, and what are the expected outcomes? While VRP is a well-known problem, providing a precise definition would improve clarity.
>
> >Sections 3.1 and 3.3: What constitutes an instance
> , and how is it encoded in the neural network? The statement that it "involves customers
> " is vague and lacks clarity.
>
> >Sections 3.1 and 3.3: How is the solution
>  defined? What features are used to describe a solution in neural networks? This information should be included, at least in the appendix.
>
>  Thank you for pointing this out. We absolutely agree that these points are currently not clear enough in the paper. We are currently improving the paper and we will upload a new version that addresses these points in the next days.
>
>  > Configuration: The paper provides detailed configuration settings but seems not clearly state how these were chosen (correct me if I am wrong). Are all the configurations tuned to the best?
>
> We do not perform any automated hyperparameter tuning for any of the approaches, including NDS. We note for NDS in the paper that the "configuration was manually
> selected rather than derived from automated hyperparameter tuning
> " on page 8. The reason for this is that automated tuning is too computationally expensive in a GPU-based setting.
>
> > Robustness to Configuration Changes: If configuration parameters are perturbed, how does the robustness of the traditional OR method compare to the proposed learning-based method?
>
> We did not notice any significant difference between traditional OR methods and NDS with respect to the robustness to configuration changes. In our experience, HGS and SISRs are especially robust to changes in their configuration. To better analyze the robustness of NDS, we will conduct some new experiments to evaluate the impact of the number of augmentations, the number of rollouts, and the threshold factor. We will update the paper once the experiments are concluded.

---

> > ### Comment · Reviewer_LxMb · 2025-02-28
> >
> > Thanks for the authors' response. After reading the revised paper, I think my concerns have been addressed.

---

### Review · Reviewer_QeAu · 2025-02-12

**Summary Of Contributions:**

This paper proposes a novel learning-based approach to solving vehicle routing problems. Unlike existing deep reinforcement learning (DRL) methods that iteratively construct solutions, the proposed approach trains a DRL agent to deconstruct a given solution by removing certain customer nodes, followed by a greedy heuristic that reinserts the removed nodes. This represents a fundamentally different strategy, as the focus of this algorithm is on efficiently exploring the solution space (like many meta-heuristics) rather than generating high-quality solutions in a few iterations (like existing DRL methods).

Overall, the paper is easy to follow. I reviewed it critically but did not identify any major flaws. I believe this paper is a strong fit for TMLR. The authors are encouraged to consider the following points to further improve the paper.

**Audience:**

Yes

**Broader Impact Concerns:**

No concerns.

**Claims And Evidence:**

Yes

**Requested Changes:**

- Clarify the questions listed above
- Potentially add ablation and numerical studies as suggested above. This is not necessary tho as the current experiments are already pretty comprehensive.

**Strengths And Weaknesses:**

**Strengths**: see summary.

**Weaknesses**:

- **Search efficiency.** The authors mention in the introduction that existing DRL methods fail to generate as many feasible solutions as state-of-the-art heuristics. While I appreciate this observation, I am not entirely sure why this is the case. For example, existing DRL agents can also explore different solutions, and this process can be accelerated with GPUs and further enhanced by more sophisticated search techniques, e.g., beam search. Since this is a key observation in the paper, I believe the limitations of existing methods warrant a more detailed discussion. Specifically, why can’t current DRL approaches efficiently explore the solution space? Could the proposed method be used to enhance their search efficiency? Moreover, why is the number of feasible solutions explored more critical than solution quality? The final point has been empirically validated, but the argument could be further strengthened by providing additional intuition on the underlying reasons.

- **Greedy insertion.** On my first read, I was initially confused by the fact that the insertion algorithm only reinserts removed nodes into the same route. For instance, in a CVRP instance with two vehicles, nodes removed from the first vehicle’s route would not be inserted into the second vehicle’s route. However, I later realized that in RL-generated VRP solutions, the routes of multiple vehicles are typically concatenated into a single array, separated by the depot node. As a result, performing removal and insertion within this concatenated array effectively enables node swapping between different routes. Is this understanding correct? The authors may want to clarify this point in the paper.

- **Initial improvement steps.** In Algorithm 2, the authors apply a few local search steps to improve the initial solution quality before calling/training the RL agent. I wonder how crucial this step is. It would be helpful if the authors provided additional ablation studies to assess its importance, especially since it adds to the computational complexity---particularly given that such steps are typically performed on CPUs, whereas other RL steps run on GPUs.

- **Generalization.** I appreciate the experimental results demonstrating the model's ability to generalize across different data distributions. Another generalization study the authors might consider is evaluating how a model trained on small instances performs on larger instances. This could be of greater practical interest, as customer distribution and vehicle capacities are typically stable, if not fixed, within a city, while the number of customers can vary significantly over time. Since logistics companies often lack the resources to train a separate model for each $n$, it would be valuable to understand how well the model generalizes in this scenario.

- **Table 2(a)**. No number is bold in the ``CVRP'' column. What do the numbers represent? Are they objective values?

---

> ### Author Response · Authors · 2025-02-18
>
> Thank you for your prompt review of our paper and for your valuable comments and suggestions. We truly appreciate your time and insights! Note that we will upload a revised version of the manuscript that includes new experiments and improvements in the coming days.
>
> We are happy that you do not have any major concerns and that you consider that paper a strong fit for TMLR. Please let us address your remaining concerns!
>
> > Search efficiency. The authors mention in the introduction that existing DRL methods fail to generate as many feasible solutions as state-of-the-art heuristics. While I appreciate this observation, I am not entirely sure why this is the case. For example, existing DRL agents can also explore different solutions, and this process can be accelerated with GPUs and further enhanced by more sophisticated search techniques, e.g., beam search. Since this is a key observation in the paper, I believe the limitations of existing methods warrant a more detailed discussion. Specifically, why can’t current DRL approaches efficiently explore the solution space? Could the proposed method be used to enhance their search efficiency? Moreover, why is the number of feasible solutions explored more critical than solution quality? The final point has been empirically validated, but the argument could be further strengthened by providing additional intuition on the underlying reasons.
>
> There are two main reasons why NDS can explore more solutions per second compared to construction approaches like POMO:
>
> 1) NDS is an improvement method rather than a construction method.
> 2) NDS is designed for efficient GPU and CPU interaction.
>
> Let us elaborate on these points:
>
> ### 1) Improvement vs. Construction
> To explore a new solution for an instance, NDS selects M customers for deconstruction and then repairs the instance. This means NDS requires only M calls to the decoder (which has an attention mechanism with O(N^2) complexity). In contrast, a construction method like POMO builds a solution from scratch, requiring N calls to the decoder with the same complexity. In our experiments, we set M = 15, while N can be as large as 2000, which explains the significant performance difference.
>
> However, while NDS involves significantly fewer decoder calls, it does require solution reconstruction using greedy insertion. Although greedy insertion is not computationally expensive, it can only be effectively executed on the CPU, and transferring data between the GPU and CPU is relatively costly.
>
> ### 2) Efficient GPU-CPU Interaction
> To mitigate the overhead of data transfer, we use an efficient implementation based on batched rollouts. This means that we compute K deconstruction operations on the GPU at once and then apply them consecutively on the CPU. Batched rollouts improve GPU utilization and reduce the frequency of data transfers between the CPU and GPU.
>
> Overall, this is a crucial aspect of our approach, and we will clarify it further in the updated version of the paper.
>
> Regarding potential improvements to learning-based construction methods, we do not believe these insights can be directly applied to enhance their performance. However, the core idea of using diverse batched rollouts for efficiency could be beneficial for other learning-based improvement methods.
>
> We do not have a theoretical explanation for why solution generation speed is particularly helpful in this context. In general, there is a trade-off between solution quality and speed, and our goal is to evaluate as many high-quality solutions per second as possible.
>
>
> > Greedy insertion. On my first read, I was initially confused by the fact that the insertion algorithm only reinserts removed nodes into the same route. For instance, in a CVRP instance with two vehicles, nodes removed from the first vehicle’s route would not be inserted into the second vehicle’s route. However, I later realized that in RL-generated VRP solutions, the routes of multiple vehicles are typically concatenated into a single array, separated by the depot node. As a result, performing removal and insertion within this concatenated array effectively enables node swapping between different routes. Is this understanding correct? The authors may want to clarify this point in the paper.
>
> Yes, your understand is correct. We will try to make this more clear.
>
>
> > Initial improvement steps. In Algorithm 2, the authors apply a few local search steps to improve the initial solution quality before calling/training the RL agent. I wonder how crucial this step is. It would be helpful if the authors provided additional ablation studies to assess its importance, especially since it adds to the computational complexity---particularly given that such steps are typically performed on CPUs, whereas other RL steps run on GPUs.
>
>
> Thank you for your suggestion. We will conduct an ablation experiment and report the results in an updated version of the paper in the next days.

---

> > ### Author Response · Authors · 2025-02-18
> >
> > > Generalization. I appreciate the experimental results demonstrating the model's ability to generalize across different data distributions. Another generalization study the authors might consider is evaluating how a model trained on small instances performs on larger instances. This could be of greater practical interest, as customer distribution and vehicle capacities are typically stable, if not fixed, within a city, while the number of customers can vary significantly over time. Since logistics companies often lack the resources to train a separate model for each
> > , it would be valuable to understand how well the model generalizes in this scenario
> >
> > This experiment has also been requested by Reviewer QeAu. It will be included in the updated version of the paper.
> >
> > > Table 2(a). No number is bold in the ``CVRP'' column. What do the numbers represent? Are they objective values?
> >
> > Yes, these are the objective values. We will include this in the table title.

---

> > > ### Comment · Reviewer_QeAu · 2025-02-28
> > >
> > > Thank you for addressing my concerns. Overall, I lean toward acceptance because this paper presents a novel idea that outperforms most learning-based methods. The modest improvement margin seems reasonable given that TMLR prioritizes the alignment of claims with evidence rather than impact. That said, I acknowledge the other reviewer’s point that the performance gains are relatively small. If the paper were ultimately rejected, I would understand the decision.
> > >
> > > I also believe that the new generalization study could really elevate the paper (if the results are strong). Many learning-based methods struggle with generalization, and I see potential in this method in that regard. However, without seeing the results, I cannot be more definitive in my assessment.

---

> > > > ### Author Response · Authors · 2025-02-28
> > > >
> > > > Thank you for your response! We have just uploaded a new version of the paper, which includes the requested generalization experiments along with many other updates. Conducting these experiments took some time, which is why the updated version of the paper was uploaded later than expected. We hope for your understanding.
> > > >
> > > > We are happy to report that the newly added generalization experiment demonstrates that NDS generalizes extremely well to instances of different sizes, with only a minor drop in performance compared to size-specific models.

---

### Review · Reviewer_tv5y · 2025-02-13

**Summary Of Contributions:**

This paper explores learning-based solution construction for VRPs and their variants, proposing a deconstruction-then-reconstruction approach. It leverages guidance from trained DNNs and takes advantage of parallel computation via GPUs. While the individual elements of the approach are simple and not new, their combination seems effective. However, the proposed method only demonstrates minor performance improvements over the baseline algorithm SISRs on every dataset.

**Audience:**

Yes

**Claims And Evidence:**

Yes

**Requested Changes:**

Below are some suggested changes:

- Add clarifications (if any), regarding the major weaknesses in the manuscript.
- Specify the static instance information mentioned in Section 3.3.1 and explain how it is encoded.

**Strengths And Weaknesses:**

### Strengths

- The paper is well-written and follows a logical structure.

### Weaknesses

- **Generally speaking, the performance gap between the proposed method and SISRs (which seems to be the best algorithm for three types of problems) is very mild (almost negligible), especially when $N$ is large. The authors should have reported the comparison results of their method against the best one (i.e., SISRs).**

My other concerns revolve around the fairness of some comparisons:

- The authors note that some approaches process instances in batches, but not address the potential impact of batch size on resource allocation when time limits are imposed. With a fixed time limit, increasing the batch size may lead to fewer resources being allocated to each individual instance, potentially resulting in lower average performance. Moreover, the batch sizes used in different approaches are inconsistent, e.g., 1 for BQ and 4/6 for UDC, as detailed in Appendix A. I suggest the authors provide further clarification on this matter.

- If I understand correctly, the baselines BQ and LEHD are trained using problem instances of size 100 and evaluated across four different problem sizes. In contrast, NDS is trained and evaluated for each problem size separately, rather than being trained on a specific size and evaluated across all sizes. For a fair comparison, BQ and LEHD should also be trained and evaluated for different problem sizes. The authors justify not doing so by claiming that these methods require near-optimal solutions for supervised learning, which are not obtainable for large problems. However, this reasoning is not entirely convincing. Off-the-shelf solvers and tools, such as SISRs, could be used to search for high-quality solutions as training labels ("near-optimal" may not be necessary). Besides, I also recommend reporting an additional NDS experiment that evaluates on the four problem sizes using the model trained on problem size 100.

Additionally, a minor concern is the lack of analysis on the impact of certain hyper-parameters in the ASA algorithm, such as the number of augmentations and rollouts. This would help readers better understand the roles of these components within the overall algorithm.

---

> ### Author Response · Authors · 2025-02-18
>
> Thank you for your prompt review of our paper and for your valuable comments and suggestions. We truly appreciate your time and insights! Note that we will upload a revised version of the manuscript that includes new experiments and improvements in the coming days.
>
> We are happy that you find our paper to be well written and that you believe that the TMLR audience will be interested in our findings. Please let us address your remaining concerns!
>
> > Generally speaking, the performance gap between the proposed method and SISRs (which seems to be the best algorithm for three types of problems) is very mild (almost negligible), especially when N
>  is large. The authors should have reported the comparison results of their method against the best one (i.e., SISRs).
>
> **Performance difference** Yes, our approach performs only slightly better than SISRs (especially for larger N). However, we find it remarkable that a learning-based approach, without relying on complex handcrafted heuristics, can achieve this level of performance. To the best of our knowledge, NDS is the first learning-based method to outperform state-of-the-art traditional techniques.
>
> **Comparison to SISRs** We provide results for SISRs across all instance sets (Table 1) and include a detailed comparison between SISRs and NDS across different runtime limits in Appendix D. The performance gap in Table 1 is measured relative to HGS, as nearly all state-of-the-art ML papers reporting results on the CVRP use HGS as their primary baseline. Following this convention ensures better comparability across different studies. For the VRPTW and PCVRP, we also use HGS (implemented in the PyVRP package) for consistency. We will make our motivation for using HGS as the baseline more clear in the paper. Please let us know if this addresses your concern.
>
> > The authors note that some approaches process instances in batches, but not address the potential impact of batch size on resource allocation when time limits are imposed. With a fixed time limit, increasing the batch size may lead to fewer resources being allocated to each individual instance, potentially resulting in lower average performance. Moreover, the batch sizes used in different approaches are inconsistent, e.g., 1 for BQ and 4/6 for UDC, as detailed in Appendix A. I suggest the authors provide further clarification on this matter.
>
> For approaches that process instances in batches, we effectively limit the runtime to batch_size × runtime_per_instance. For example, this means all approaches are given approximately 50,000 seconds to solve the entire CVRP100 test set. This setup actually gives ML approaches that process instances in batches an advantage over methods that solve instances one by one (like our approach). We realized that our current explanation in the paper is misleading, and we will revise it for better clarity.
>
> The batch size for different ML approaches is chosen to fully utilize the available GPU memory. Naturally, approaches that generate more solutions in parallel require more GPU memory. We will ensure this is clearly explained in the updated version of the paper.

---

> > ### Author Response · Authors · 2025-02-18
> >
> > > If I understand correctly, the baselines BQ and LEHD are trained using problem instances of size 100 and evaluated across four different problem sizes. In contrast, NDS is trained and evaluated for each problem size separately, rather than being trained on a specific size and evaluated across all sizes. For a fair comparison, BQ and LEHD should also be trained and evaluated for different problem sizes. The authors justify not doing so by claiming that these methods require near-optimal solutions for supervised learning, which are not obtainable for large problems. However, this reasoning is not entirely convincing. Off-the-shelf solvers and tools, such as SISRs, could be used to search for high-quality solutions as training labels ("near-optimal" may not be necessary). Besides, I also recommend reporting an additional NDS experiment that evaluates on the four problem sizes using the model trained on problem size 100.
> >
> > **Comparison to BQ and LEHD**
> > Your understanding of the experimental setup is correct. We do not retrain BQ and LEHD on larger instances for three main reasons:
> >
> > 1. Both approaches are explicitly designed for generalization. For example, the full title of the LEHD paper is "Neural Combinatorial Optimization with Heavy Decoder: Toward Large Scale Generalization." Even in their original paper, the authors train on instances with 100 nodes and evaluate on instances with up to 1,000 for their main experiment.
> >
> > 2. Generating the required training sets would be very computationally expensive. Both approaches require 1 million training instances, and generating these with SISRs under our default runtime limits would require approximately 120,000 CPU hours.
> >
> > 3. There is no strong reason to believe that retraining LEHD and BQ would make them competitive with our approach. Even on CVRP100, where the test set matches the training distribution, both approaches perform significantly worse despite being trained on near-optimal solutions.
> >
> > We hope these points clarify our decision and convince you that our comparison to BQ and LEHD is fair.
> >
> > **Additional generalization experiments** We will gladly perform the additional generalization experiments of NDS that you suggested. We will update the paper once the experiments are finished!
> >
> >
> > > Additionally, a minor concern is the lack of analysis on the impact of certain hyper-parameters in the ASA algorithm, such as the number of augmentations and rollouts. This would help readers better understand the roles of these components within the overall algorithm.
> >
> > We agree with your assessment. We will conduct some experiments to evaluate the impact of the number of augmentations, the number of rollouts, and the threshold factor.
> >
> > > Below are some suggested changes:
> > Add clarifications (if any), regarding the major weaknesses in the manuscript.
> > Specify the static instance information mentioned in Section 3.3.1 and explain how it is encoded.
> >
> > Thank you for your suggestions. We will make the requested changes.

---

> > ### Comment · Reviewer_tv5y · 2025-02-28
> >
> > Thanks for your reply. Since the authors also acknowledge that the performance improvement of the proposed approach over SISRs is very mild, I think the contribution of this work is minor and lean towards rejection.
> >
> > I think Table 1 is quite misleading and strongly suggest that the comparison results of the proposed approach against the best one (e.g., SISRs) be presented in the main text.

---

> > > ### Author Response · Authors · 2025-02-28
> > >
> > > Thank you for your response! We have uploaded a new version of the paper that includes your an evaluation of the generalization capabilities of NDS as well as an analysis of the impact of certain hyperparameters in the ASA algorithm. Conducting these experiments took some time, which is why the updated version of the paper was uploaded so late. We hope for your understanding.
> > >
> > > Please allow us to address your two remaining concerns.
> > >
> > > **Performance in comparison to SISRs**
> > > While the absolute performance difference between NDS and SISRs may seem mild, the relative difference remains significant. Figure 3 (and Figure 9 in the Appendix) provides a direct performance comparison between NDS and SISRs. For example, on the CVRP500 instances, NDS requires just 25 seconds to match the performance that SISRs achieves in 250 seconds.
> > >
> > > **Table 1**
> > > Could you clarify why you believe Table 1 is misleading? Is your only concern that the gap is calculated with respect to HGS rather than SISRs? We have provided reasons for calculating the gap with respect to HGS in our initial response and have made this clearer in the updated version of the paper. Additionally, Figures 3 and 9 provide a detailed comparison between NDS and SISRs, demonstrating that we do not shy away from a direct comparison between NDS and SISRs.

---

### Author Response · Authors · 2025-02-28

Thank you all once again for your valuable feedback. Conducting the additional experiments took some time, but we have now finally uploaded a revised version of the paper that incorporates your suggestions and comments. We apologize for the delay.

In this updated version, all changes are highlighted in red.


Below is a summary of the key revisions:

- **New generalization experiment:** We have added an evaluation of NDS's ability to generalize to both smaller and larger instances (Appendix G). Our findings show that NDS generalizes exceptionally well across different instance sizes.
- **New sensitivity analysis:** We conducted an additional experiment to assess the impact of key hyperparameters on search performance (Appendix F). Results indicate that NDS is robust to hyperparameter choices, suggesting that hyperparameter tuning is not essential for strong performance.
- **New ablation experiment:** We conducted an ablation study that removes the initial solution improvement step during training (Appendix F). The results highlight its critical role in achieving strong performance.

- A new section, *Vehicle Routing Problems*, has been added to provide a more detailed explanation of the three considered problems.
- We have included a high-level overview of NDS at the beginning of Section 4, along with an explanation of why NDS is faster than construction methods.
- The introduction of Section 3.3.1 has been improved and expanded to better describe the model input and its processing.
- We have clarified our motivation for selecting HGS as the main baseline in Section 5.2.
- The explanation of the time limit for approaches that process instances in batches has been corrected in Section 5.2.
- Additional details on the selected batch size values are now provided in Appendix A.
- In Section 4.2, we have made it more clear that the greedy insertion algorithm considers inserting a removed customer into all routes, not just the route from which it was removed.
- The conclusion now includes a more detailed discussion of the weaknesses of our approach.
- The caption of Table 2 now explicitly states that the reported values correspond to the objective function values.

We appreciate your time and thoughtful feedback, which have helped us improve the paper significantly. Please let us know if you have any more questions.

---

### Decision · Action_Editor_Nfdk · 2025-04-02

**Recommendation:** Accept with minor revision

**Comment:**

The reviewers agree that the latest version of the manuscript should be accepted. Congratulations!

One remaining concern we discussed is how the GAP is (and should be) calculated in Table 3. One reviewer found that computing it with respect to the baseline might be misleading to readers at first glimpse. While NDS does seem to dominate other methods, it appears to be fairly close to the results of some of them (e.g. SISRs). On the other hand, I surveyed a few related papers, and it seems to be a common practice in the literature.

In any case, the Obj seems to be the critical metric. With that in mind, I suggest the others focus on that metric and, for example, bold that column instead of the Gap. It would also be helpful to be explicit about what bolding means (e.g. by specifying it in the caption). If it's "simply" the highest value, it might be worth also highlighting the second-best (ideally, if you have a measure of uncertainty, bold all methods that are statistically indistinguishable).

In the caption, you mention a 7% gap, but as far as I understand, it's closer to 5-6% ( (59.45 - 56)/59.45 ), no? Maybe I am misunderstanding.

Please update the caption to reflect the changes you end up making.

**Audience:**

The reviewers all agree that some individuals in the TMLR audience would be interested in this work.

**Claims And Evidence:**

Yes, the reviewers agree that the claims are well supported. I discuss one somewhat minor point below that I believe is easy to fix.

---

> ### Author Response · Authors · 2025-05-04
>
> Dear Action Editor,
>
> Thank you for your thoughtful feedback and for overseeing the review process. We are grateful to you and the reviewers for your time and constructive suggestions.
>
> We have uploaded the camera-ready version of the manuscript which incorporates all the requested changes. In particular, we have updated Table 1 and its caption to emphasize the objective value instead of the gap values.
> Regarding the ~7% gap between NDS and LEHD, we note that it is calculated based on LEHD’s performance when given the same runtime as NDS (i.e., an objective value of 60.11), and not when running for 250 stages (i.e., an objective value of 59.45). We have clarified this point in the paper.
>
> Best regards,
>
> The authors